# Convergence and Alignment of Gradient Descent with Random Backpropagation Weights

**Ganlin Song**[*]     **Ruitu Xu**[*]     **John Lafferty**[*†]

[*]Department of Statistics and Data Science
[†]Wu Tsai Institute
Yale University
{ganlin.song, ruitu.xu, john.lafferty}@yale.edu

## Abstract

Stochastic gradient descent with backpropagation is the workhorse of artificial neural networks. It has long been recognized that backpropagation fails to be a biologically plausible algorithm. Fundamentally, it is a non-local procedure—updating one neuron's synaptic weights requires knowledge of synaptic weights or receptive fields of downstream neurons. This limits the use of artificial neural networks as a tool for understanding the biological principles of information processing in the brain. Lillicrap et al. (2016) propose a more biologically plausible "feedback alignment" algorithm that uses random and fixed backpropagation weights, and show promising simulations. In this paper we study the mathematical properties of the feedback alignment procedure by analyzing convergence and alignment for two-layer networks under squared error loss. In the overparameterized setting, we prove that the error converges to zero exponentially fast, and also that regularization is necessary in order for the parameters to become aligned with the random backpropagation weights. Simulations are given that are consistent with this analysis and suggest further generalizations. These results contribute to our understanding of how biologically plausible algorithms might carry out weight learning in a manner different from Hebbian learning, with performance that is comparable with the full non-local backpropagation algorithm.

## 1 Introduction

The roots of artificial neural networks draw inspiration from networks of biological neurons (Rumelhart et al., 1986a; Elman et al., 1996; Medler, 1998). Grounded in simple abstractions of membrane potentials and firing, neural networks are increasingly being employed as a computational tool for better understanding the biological principles of information processing in the brain; examples include Yildirim et al. (2019) and Yamins & DiCarlo (2016). Even when full biological fidelity is not required, it can be useful to better align the computational abstraction with neuroscience principles.

Stochastic gradient descent has been a workhorse of artificial neural networks. Conveniently, calculation of gradients can be carried out using the backpropagation algorithm, where reverse mode automatic differentiation provides a powerful way of computing the derivatives for general architectures (Rumelhart et al., 1986b). Yet it has long been recognized that backpropagation fails to be a biologically plausible algorithm. Fundamentally, it is a non-local procedure—updating the weight between a presynaptic and postsynaptic neuron requires knowledge of the weights between the postsynaptic neuron and other neurons. No known biological mechanism exists for propagating

35th Conference on Neural Information Processing Systems (NeurIPS 2021).

information in this manner. This limits the use of artificial neural networks as a tool for understanding learning in the brain.

A wide range of approaches have been explored as a potential basis for learning and synaptic plasticity. Hebbian learning is the most fundamental procedure for adjusting weights, where repeated stimulation by a presynaptic neuron that results in the subsequent firing of the postsynaptic neuron will result in an increased strength in the connection between the two cells (Hebb, 1961; Paulsen & Sejnowski, 2000). Several variants of Hebbian learning, some making connections to principal components analysis, have been proposed (Oja, 1982; Sejnowski & Tesauro, 1989; Sejnowski, 1999). In this paper, our focus is on a formulation of Lillicrap et al. (2016) based on random backpropagation weights that are fixed during the learning process, called the "feedback alignment" (FA) algorithm. Lillicrap et al. (2016) show that the model can still learn from data, and observe the interesting phenomenon that the error signals propagated with the forward weights align with those propagated with fixed random backward weights during training. Direct feedback alignment (DFA) (Nøkland, 2016) extends FA by adding skip connections to send the error signals directly to each hidden layer, allowing parallelization of weight updates. Empirical studies given by Launay et al. (2020) show that DFA can be successfully applied to train a number of modern deep learning models, including transformers. Based on DFA, Frenkel et al. (2021) proposes direct the random target projection (DRTP) algorithm that trains the network weights with a random projection of the target vector instead of the error, and shows alignment for linear networks. Related proposals, including methods based on the use of differences of neuron activities, have been made in a series of recent papers (Akrout et al., 2019; Bellec et al., 2019; Lillicrap et al., 2020). A comparison of some of these methods is made by Bartunov et al. (2018).

The use of random feedback weights, which are not directly tied to the forward weights, removes issues of non-locality. However, it is not clear under what conditions optimization of error and learning can be successful. While Lillicrap et al. (2016) give suggestive simulations and some analysis for the linear case, it has been an open problem to explain the behavior of this algorithm for training the weights of a neural network. In this paper, we study the mathematical properties of the feedback alignment procedure by analyzing convergence and alignment for two-layer networks under squared error loss. In the overparameterized setting, we prove that the error converges to zero exponentially fast. We also show, unexpectedly, that the parameters become aligned with the random backpropagation weights only when regularization is used. Simulations are given that are consistent with this analysis and suggest further generalizations. The following section gives further background and an overview of our results.

## 2    Problem Statement and Overview of Results

In this section we provide a formulation of the backpropagation algorithm to establish notation and the context for our analysis. We then formulate the feedback aligment algorithm that uses random backpropation weights. A high-level overview of our results is then presented, together with some of the intuition and proof techniques behind these results; we also contrast with what was known previously.

We mainly consider two-layer neural networks in the regression setting, specified by a family of functions $f : \mathbb{R}^d \to \mathbb{R}$ with input dimension $d$, sample size $n$, and $p$ neurons in the hidden layer. For an input $x \in \mathbb{R}^d$, the network outputs

$$f(x) = \frac{1}{\sqrt{p}} \sum_{r=1}^{p} \beta_r \psi(w_r^\intercal x) = \frac{1}{\sqrt{p}} \beta^\intercal \psi(Wx), \tag{2.1}$$

where $W = (w_1, ..., w_p)^\intercal \in \mathbb{R}^{p \times d}$ and $\beta = (\beta_1, ..., \beta_p)^\intercal \in \mathbb{R}^p$ represent the feed-forward weights in the first and second layers, and $\psi$ denotes an element-wise activation function. The scaling by $\sqrt{p}$ is simply for convenience in the analysis.

Given $n$ input-response pairs $\{(x_i, y_i)\}_{i=1}^n$, the training objective is to minimize the squared error

$$\mathcal{L}(W, \beta) = \frac{1}{2} \sum_{i=1}^{n} \left( y_i - f(x_i) \right)^2. \tag{2.2}$$

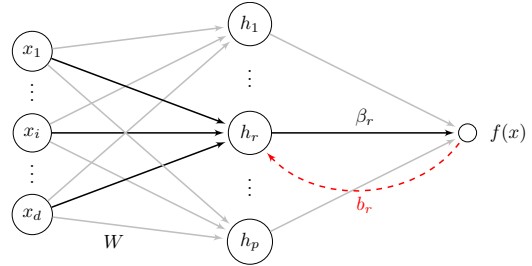

**Algorithm 1** Feedback Alignment

**Input:** Dataset $\{(x_i, y_i)\}_{i=1}^n$, step size $\eta$
1: **initialize** $W$, $\beta$ and $b$ as Gaussian
2: **while** not converged **do**
3:     $\beta_r \leftarrow \beta_r - \frac{\eta}{\sqrt{p}} \sum_{i=1}^n e_i \psi(w_r^\mathsf{T} x_i)$
4:     $w_r \leftarrow w_r - \frac{\eta}{\sqrt{p}} \sum_{i=1}^n e_i b_r \psi'(w_r^\mathsf{T} x_i) x_i$
5:     for $r \in [p]$
6: **end while**

Figure 1: Standard backpropagation updates the first layer weights for a hidden node $r$ with the second layer feedforward weight $\beta_r$. We study the procedure where the error is backpropagated instead using a fixed, random weight $b_r$.

Standard gradient descent attempts to minimize (2.2) by updating the feed-forward weights following gradient directions according to

$$\beta_r(t+1) = \beta_r(t) - \eta \frac{\partial \mathcal{L}}{\partial \beta_r}(W(t), \beta(t))$$

$$w_r(t+1) = w_r(t) - \eta \frac{\partial \mathcal{L}}{\partial w_r}(W(t), \beta(t)),$$

for each $r \in [p]$, where $\eta > 0$ denotes the step size. We initialize $\beta(0)$ and $w_r(0)$ as standard Gaussian vectors. We introduce the notation $f(t), e(t) \in \mathbb{R}^n$, with $f_i(t) = f(x_i)$ denoting the network output on input $x_i$ when the weights are $W(t)$ and $\beta(t)$, and $e_i(t) = y_i - f_i(t)$ denoting the corresponding prediction error or residual. With this notation, the gradients are expressed as

$$\frac{\partial \mathcal{L}}{\partial \beta_r} = \frac{1}{\sqrt{p}} \sum_{i=1}^n e_i \psi(w_r^\mathsf{T} x_i), \quad \frac{\partial \mathcal{L}}{\partial w_r} = \frac{1}{\sqrt{p}} \sum_{i=1}^n e_i \beta_r \psi'(w_r^\mathsf{T} x_i) x_i.$$

Here it is seen that the the gradient of the first-layer weights $\frac{\partial \mathcal{L}}{\partial w_r}$ involves not only the local input $x_i$ and the change in the response of the $r$-th neuron, but also the backpropagated error signal $e_i \beta_r$. The appearance of $\beta_r$ is, of course, due to the chain rule; but in effect it requires that the forward weights between layers are identical to the backward weights under error propagation. There is no evidence of biological mechanisms that would enable such "synaptic symmetry."

In the *feedback alignment* procedure of (Lillicrap et al., 2016), when updating the weights $w_r$, the error signal is weighted, and propagated backward, not by the second layer feedforward weights $\beta$, but rather by a random set of weights $b \in \mathbb{R}^p$ that are fixed during the course of training. Equivalently, the gradients for the first layer are replaced by the terms

$$\widetilde{\frac{\partial \mathcal{L}}{\partial w_r}} = \frac{1}{\sqrt{p}} \sum_{i=1}^n e_i b_r \psi'(w_r^\mathsf{T} x_i) x_i. \tag{2.3}$$

Note, however, that this update rule does not correspond to the gradient with respect to a modified loss function. The use of a random weight $b_r$ when updating the first layer weights $w_r$ does not violate locality, and could conceivably be implemented by biological mechanisms; we refer to Lillicrap et al. (2016); Bartunov et al. (2018); Lillicrap et al. (2020) for further discussion. A schematic of the relationship between the two algorithms is shown in Figure 1.

We can now summarize the main results and contributions of this paper. Our first result shows that the error converges to zero when using random backpropagation weights.

- Under Gaussian initialization of the parameters, if the model is sufficiently over-parameterized with $p \gg n$, then the error converges to zero linearly. Moreover, the parameters satisfy $\|w_r(t) - w_r(0)\| = \widetilde{O}\left(\frac{n}{\sqrt{p}}\right)$ and $|\beta_r(t) - \beta_r(0)| = \widetilde{O}\left(\frac{n}{\sqrt{p}}\right)$.

The precise assumptions and statement of this result are given in Theorem 3.2. The proof shows in the over-parameterized regime that the weights only change by a small amount. While related to

results for standard gradient descent, new methods are required because the "effective kernel" is not positive semi-definite.

We next turn to the issue of alignment of the second layer parameters $\beta$ with the random back-propagation weights $b$. Such alignment was first observed in the original simulations of Lillicrap et al. (2016). With $h \in \mathbb{R}^p$ denoting the hidden layer of the two-layer network, the term $\delta_{\mathrm{BP}}(h) := \frac{\partial \mathcal{L}}{\partial h} = \frac{1}{\sqrt{p}} \beta \sum_{i=1}^n e_i$ represents how the error signals $e_i$ are sent backward to update the feed-forward weights. With the use of random backpropagation weights, the error is instead propagated backward as $\delta_{\mathrm{FA}}(h) = \frac{1}{\sqrt{p}} b \sum_{i=1}^n e_i$.

Lillicrap et al. (2016) notice a decreasing angle between $\delta_{\mathrm{BP}}(h)$ and $\delta_{\mathrm{FA}}(h)$ during training, which is a sufficient condition to ensure that the algorithm converges. In the case of $k$-way classification, the last layer has $k$ nodes, $\beta$ and $b$ are $p \times k$ matrices, and each error term $e_i$ is a $k$-vector. In the regression setting, $k = 1$ so the angle between $\delta_{\mathrm{BP}}(h)$ and $\delta_{\mathrm{FA}}(h)$ is the same as the angle between $\beta$ and $b$. Intuitively, the possibility for alignment is seen in the fact that while the updates for $W$ use the error weighted by the random weights $b$, the updates for $\beta$ indirectly involve $W$, allowing for the possibility that dependence on $b$ will be introduced into $\beta$.

Our first result shows that, in fact, alignment will *not* occur in the over-parameterized setting. (So, while the error may still converge, "feedback alignment" may be a bit of a misnomer for the algorithm.)

- The cosine of the angle between the $p$-dimensional vectors $\delta_{\mathrm{FA}}$ and $\delta_{\mathrm{BP}}$ satisfies $\cos \angle(\delta_{\mathrm{FA}}, \delta_{\mathrm{BP}}(t)) = \cos \angle(b, \beta(t)) = O\left(\frac{n}{\sqrt{p}}\right)$.

However, we show that regularizing the parameters will cause $\delta_{\mathrm{BP}}$ to align with $\delta_{\mathrm{FA}}$ and therefore the parameters $\beta$ to align with $b$. Since $\beta(0)$ and $b$ are high dimensional Gaussian vectors, they are nearly orthogonal with high probability. The effect of regularization can be seen as shrinking the component of $\beta(0)$ in the parameters over time. Our next result establishes this precisely in the linear case.

- Supposing that $\psi(u) = u$, then introducing a ridge penalty $\lambda(t)\|\beta\|^2$ where $\lambda(t) = \lambda$ for $t \leq T$ and $\lambda(t) = 0$ for $t > T$ on $\beta$ causes the parameters to align, with $\cos \angle(b, \beta(t)) \geq c > 0$ for sufficiently large $t$.

The technical conditions are given in Theorem 4.6. Our simulations are consistent with this result, and also show alignment with a constant regularization $\lambda(t) \equiv \lambda$, for both linear and nonlinear activation functions. Finally, we complement this result by showing that convergence is preserved with regularization, for general activation functions. This is presented in Theorem 4.2.

## 3 Convergence with Random Backpropagation Weights

Due to the replacement of backward weights with the random backpropagation weights, there is no guarantee *a priori* that the algorithm will reduce the squared error loss $\mathcal{L}$. Lillicrap et al. (2020) study the convergence on two-layer linear networks in a continuous time setting. Through the analysis of a system of differential equations on the network parameters, convergence to the true linear target function is shown, in the population setting of arbitrarily large training data. Among recent studies of over-parametrized networks under backpropagation, the neural tangent kernel (NTK) is heavily utilized to describe the evolution of the network during training (Jacot et al., 2018; Chen & Xu, 2020). For any neural network $f(x, \theta)$ with parameter $\theta$, the NTK is defined as

$$K_f(x, y) = \left\langle \frac{\partial f(x, \theta)}{\partial \theta}, \frac{\partial f(y, \theta)}{\partial \theta} \right\rangle.$$

Given a dataset $\{(x_i, y_i)\}_{i=1}^n$, we can also consider its corresponding Gram matrix $K = (K_f(x_i, x_j))_{n \times n}$. Jacot et al. (2018) show that in the infinite width limit, $K_f$ converges to a constant at initialization and does not drift away from initialization throughout training. In the over-parameterized setting, if the Gram matrix $K$ is positive definite, then $K$ will remain close to its initialization during training, resulting in linear convergence of the squared error loss (Du et al., 2018, 2019; Gao & Lafferty, 2020). For the two-layer network $f(x, \theta)$ defined in (2.1) with $\theta = (\beta, W)$,

the kernel $K_f$ can be written in two parts, $G_f$ and $H_f$, which correspond to $\beta$ and $W$ respectively:

$$K_f(x,y) = G_f(x,y) + H_f(x,y) := \left\langle \frac{\partial f(x,\theta)}{\partial \beta}, \frac{\partial f(y,\theta)}{\partial \beta} \right\rangle + \sum_{r=1}^{p} \left\langle \frac{\partial f(x,\theta)}{\partial w_r}, \frac{\partial f(y,\theta)}{\partial w_r} \right\rangle.$$

Under the feedback alignment scheme with random backward weights $b$, $G_f$ remains the same as for standard backpropagation, while one of the gradient terms $\frac{\partial f}{\partial w_r}$ in $H_f$ changes to $\widetilde{\frac{\partial f(x,\theta)}{\partial w_r}} = \frac{1}{\sqrt{p}} b_r \psi'(w_r^\mathsf{T} x) x$, with $H_f$ replaced by $H_f = \sum_{r=1}^{p} \left\langle \widetilde{\frac{\partial f(x,\theta)}{\partial w_r}}, \frac{\partial f(y,\theta)}{\partial w_r} \right\rangle$. As a result, $H_f$ is no longer positive semi-definite and close to 0 at initialization if the network is over-parameterized. However, if $G = (G_f(x_i,x_j))_{n\times n}$ is positive definite and $H = (H_f(x_i,x_j))_{n\times n}$ remains small during training, we are still able to show that the loss $\mathcal{L}$ will converge to zero exponentially fast.

**Assumption 3.1.** Define the matrix $\overline{G} \in \mathbb{R}^{n\times n}$ with entries $\overline{G}_{i,j} = \mathbb{E}_{w\sim\mathcal{N}(0,I_p)}\psi(w^\mathsf{T} x_i)\psi(w^\mathsf{T} x_j)$. Then we assume that the minimum eigenvalue satisfies $\lambda_{\min}(\overline{G}) \geq \gamma$, where $\gamma$ is a positive constant.

**Theorem 3.2.** *Let $W(0)$, $\beta(0)$ and $b$ have i.i.d. standard Gaussian entries. Assume (1) Assumption 3.1 holds, (2) $\psi$ is smooth, $\psi$, $\psi'$ and $\psi''$ are bounded and (3) $|y_i|$ and $\|x_i\|$ are bounded for all $i \in [n]$. Then there exists positive constants $c_1$, $c_2$, $C_1$ and $C_2$, such that for any $\delta \in (0,1)$, if $p \geq \max\left(C_1 \frac{n^2}{\delta\gamma^2}, C_2 \frac{n^4 \log p}{\gamma^4}\right)$, then with probability at least $1-\delta$ we have that*

$$\|e(t+1)\| \leq (1 - \frac{\eta\gamma}{4})\|e(t)\| \tag{3.1}$$

*and*

$$\|w_r(t) - w_r(0)\| \leq c_1 \frac{n\sqrt{\log p}}{\gamma\sqrt{p}}, \quad |\beta_r(t) - \beta_r(0)| \leq c_2 \frac{n}{\gamma\sqrt{p}} \tag{3.2}$$

*for all $r \in [p]$ and $t > 0$.*

We note that the matrix $\overline{G}$ in Assumption 3.1 is the expectation of $G$ with respect to the random initialization, and is thus close to $\overline{G}$ due to concentration. To justify the assumption, we provide the following proposition, which states that Assumption 3.1 holds when the inputs $x_i$ are drawn independently from a Gaussian distribution. The proofs of Theorem 3.2 and Proposition 3.3 are deferred to **??**.

**Proposition 3.3.** *Suppose $x_1, ..., x_n \overset{i.i.d.}{\sim} \mathcal{N}(0, I_d/d)$ and the activation function $\psi$ is sigmoid or tanh. If $d = \Omega(n)$, then Assumption 3.1 holds with high probability.*

## 4 Alignment with Random Backpropagation Weights

The most prominent characteristic of the feedback alignment algorithm is the phenomenon that the error signals propagated with the forward weights align with those propagated with fixed random backward weights during training. Specifically, if we denote $h \in \mathbb{R}^p$ to be the hidden layer of the network, then we write $\delta_{\mathrm{BP}}(h) := \frac{\partial \mathcal{L}}{\partial h}$ to represent the error signals with respect to the hidden layer that are backpropagated with the feed-forward weights and $\delta_{\mathrm{FA}}(h)$ as the error signals computed with fixed random backward weights. In particular, the error signals $\delta_{\mathrm{BP}}(h)$ and $\delta_{\mathrm{FA}}(h)$ for the two-layer network (2.1) are given by

$$\delta_{\mathrm{BP}}(h) = \frac{1}{\sqrt{p}}\beta \sum_{i=1}^{n} e_i \quad \text{and} \quad \delta_{\mathrm{FA}}(h) = \frac{1}{\sqrt{p}}b \sum_{i=1}^{n} e_i.$$

Lillicrap et al. (2016) notice a decreasing angle between $\delta_{\mathrm{BP}}(h)$ and $\delta_{\mathrm{FA}}(h)$ during training. We formalize this concept of alignment by the following definition.

**Definition 4.1.** We say a two-layer network *aligns* with the random weights $b$ during training if there exists a constant $c > 0$ and time $T_c$ such that $\cos\angle(\delta_{\mathrm{FA}}, \delta_{\mathrm{BP}}(t)) = \cos\angle(b, \beta(t)) = \frac{\langle b, \beta(t)\rangle}{\|b\|\|\beta(t)\|} \geq c$ for all $t > T_c$.

## 4.1 Regularized feedback alignment

Unfortunately, alignment between $\beta(t)$ and $b$ is not guaranteed for over-parameterized networks and the loss (2.2). In particular, we control the cosine value of the angle by inequalities (3.2) from Theorem 3.2, *i.e.*,

$$\left| \cos \angle (b, \beta(t)) \right| \leq \frac{|\langle \frac{b}{\|b\|}, \beta(0) \rangle| + \|\beta(t) - \beta(0)\|}{\|\beta(0)\| - \|\beta(t) - \beta(0)\|} = O\left(\frac{n}{\sqrt{p}}\right),$$

which indicates that $\beta(t)$ and $b$ become orthogonal as the network becomes wider. Intuitively, this can be understood as resulting from the parameters staying near their initializations during training when $p$ is large, where $\beta(0)$ and $b$ are almost orthogonal to each other. This motivates us to regularize the network parameters. We consider in this work the squared error loss with an $\ell_2$ regularization term on $\beta$:

$$\mathcal{L}(t, W, \beta) = \frac{1}{2} \sum_{i=1}^{n} \left( f(x_i) - y_i \right)^2 + \frac{1}{2} \lambda(t) \|\beta\|^2, \tag{4.1}$$

where $\{\lambda(t)\}_{t=0}^{\infty}$ is a sequence of regularization rates, which defines a series of loss functions for different training steps $t$. Thus, the update for $w_r$ remains the same and the update for $\beta$ changes to

$$\beta_r(t+1) = (1 - \lambda(t))\beta_r(t) - \frac{\eta}{\sqrt{p}} \sum_{i=1}^{n} e_i(t)\psi(w_r(t)^\mathsf{T} x_i), \quad \text{for } r \in [p].$$

Comparing to Algorithm 1, an extra contraction factor $1 - \lambda(t)$ is added in the update of $\beta(t)$, which doesn't affect the locality of the algorithm but helps the alignment by shrinking the component of $\beta(0)$ in $\beta(t)$.

Following Theorem 3.2, we provide an error bound for regularized feedback alignment in Theorem 4.2. Since regularization terms $\lambda(t)$ make additional contributions to the error $e(t)$ as well as to the kernel matrix $G$, an upper bound on $\sum_{t \geq 0} \lambda(t)$ is needed to ensure positivity of the minimal eigenvalue of $G$ during training, in order for the error $e(t)$ to be controlled. In particular, if there is no regularization, *i.e.*, $\lambda(t) = 0$ for all $t \geq 0$, then we recover exponential convergence for the error $e(t)$ as in Theorem 3.2. The proof of Theorem 4.2 is also deferred to **??**.

**Theorem 4.2.** *Assume all the conditions from Theorem 3.2. Assume* $\sum_{t=0}^{\infty} \lambda(t) \leq \tilde{S}_\lambda = \tilde{c}_S \frac{\gamma^2 \sqrt{p}}{\eta n^2 \sqrt{\log p}}$ *for some constant* $\tilde{c}_S$. *Then there exist positive constants* $C_1$ *and* $C_2$, *such that for any* $\delta \in (0, 1)$, *if* $p \geq \max\left(C_1 \frac{n^2}{\delta \gamma^2}, C_2 \frac{n^4 \log p}{\gamma^4}\right)$, *then with probability at least* $1 - \delta$, *we have*

$$\|e(t+1)\| \leq \left(1 - \frac{\eta \gamma}{4} - \eta \lambda(t)\right) \|e(t)\| + \lambda(t)\|y\| \tag{4.2}$$

*for all* $t \geq 0$.

## 4.2 Alignment analysis for linear networks

In this section, we focus on the theoretical analysis of alignment for linear networks, which is equivalent to setting the activation function $\psi$ to the identity map. The loss function can be written as

$$\mathcal{L}(t, W, \beta) = \frac{1}{2} \left\| \frac{1}{\sqrt{p}} X W^\mathsf{T} \beta - y \right\|^2 + \frac{\lambda(t)}{2} \|\beta\|^2,$$

where $X = (x_1, \ldots, x_n)^\mathsf{T}$; this is a form of over-parameterized ridge regression. Before presenting our results on alignment, we first provide a linear version of Theorem 4.2 that adopts slightly different conditions.

**Theorem 4.3.** *Assume (1)* $\|y\| = \Theta(\sqrt{n})$, $\lambda_{\min}(XX^\mathsf{T}) > \gamma$ *and* $\lambda_{\max}(XX^\mathsf{T}) < M$ *for some constants* $M > \gamma > 0$, *and (2)* $\sum_{t=0}^{\infty} \lambda(t) \leq S_\lambda = c_S \frac{\gamma \sqrt{\gamma p}}{\eta \sqrt{nM}}$ *for some constant* $c_S$. *Then for any* $\delta \in (0, 1)$, *if* $p = \Omega(\frac{Md \log(d/\delta)}{\gamma})$, *the following inequality holds for all* $t \geq 0$ *with probability at least* $1 - \delta$:

$$\|e(t+1)\| \leq \left(1 - \frac{\eta \gamma}{2} - \eta \lambda(t)\right) \|e(t)\| + \lambda(t)\|y\|. \tag{4.3}$$

We remark that in the linear case, the kernel matrix $G$ reduces to the form $XW^\mathsf{T}WX^\mathsf{T}$ and its expectation $\overline{G}$ at initialization also reduces to $XX^\mathsf{T}$. Thus, Assumption 3.1 holds if $XX^\mathsf{T}$ is positive definite, which is equivalent to the $x_i$'s being linearly independent. The result of Theorem 4.2 can not be directly applied to the linear case since we assume that $\psi$ is bounded, which is true for sigmoid or $\tanh$ but not for the identity map. This results in a slightly different order for $S_\lambda$ and an improved order for $p$.

Our results on alignment also rely on an isometric condition on $X$, which requires the minimum and the maximum eigenvalues of $XX^\mathsf{T}$ to be sufficiently close (*cf.* Definition 4.4). On the other hand, this condition is relatively mild and can be satisfied when $X$ has random Gaussian entries with a gentle dimensional constraint, as demonstrated by Proposition 4.5. Finally, we show in Theorem 4.6 that under a simple regularization strategy where a constant regularization is adopted until a cutoff time $T$, regularized feedback alignment achieves alignment if $X$ satisfies the isometric condition.

**Definition 4.4** (($\gamma, \varepsilon$)-Isometry)**.** Given positive constants $\gamma$ and $\varepsilon$, we say $X$ is ($\gamma, \varepsilon$)-isometric if $\lambda_{\min}(XX^\mathsf{T}) \geq \gamma$ and $\lambda_{\max}(XX^\mathsf{T}) \leq (1+\varepsilon)\gamma$.

**Proposition 4.5.** *Assume $X \in \mathbb{R}^{n \times d}$ has independent entries drawn from $N(0, 1/d)$. For any $\varepsilon \in (0, 1/2)$ and $\delta \in (0, 1)$, if $d = \Omega(\frac{1}{\varepsilon}\log\frac{n}{\delta} + \frac{n}{\varepsilon}\log\frac{1}{\varepsilon})$, then $X$ is $(1-\varepsilon, 4\varepsilon)$-isometric with probability $1 - \delta$.*

**Theorem 4.6.** *Assume all conditions from Theorem 4.3 hold and $X$ is ($\gamma, \varepsilon$)-isometric with a small constant $\varepsilon$. Let the regularization weights satisfy*

$$\lambda(t) = \begin{cases} \lambda, & t \leq T, \\ 0, & t > T, \end{cases}$$

*with $\lambda = L\gamma$ and $T = \lfloor S_\lambda/\lambda \rfloor$ for some large constant $L$. Then for any $\delta \in (0, 1)$, if $p = \Omega(d\log(d/\delta))$, with probability at least $1 - \delta$, regularized feedback alignment achieves alignment. Specifically, there exist a positive constant $c = c_\delta$ and time $T_c$, such that $\cos\angle(b, \beta(t)) \geq c$ for all $t > T_c$.*

We defer the proofs of Proposition 4.5, Theorem 4.3 and Theorem 4.6 to **??**. In fact, we prove Theorem 4.6 by directly computing $\beta(t)$ and the cosine of the angle. Although $b$ doesn't show up in the update of $\beta$, it can still propagate to $\beta$ through $W$. Since the size of the component of $b$ in $\beta(t)$ depends on the inner-product $\langle e(t), e(t') \rangle$ for all previous steps $t' \leq t$, the norm bound (4.3) from Theorem 4.3 is insufficient; thus, a more careful analysis of $e(t)$ is required.

We should point out that the constant $c$ in the lower bound is independent of the sample size $n$, input dimension $d$, network width $p$ and learning rate $\eta$. We also remark that the cutoff schedule of $\lambda(t)$ is just chosen for simplicity. For other schedules such as inverse-squared decay or exponential decay, one could also obtain the same alignment result as long as the summation of $\lambda(t)$ is less than $S_\lambda$.

**Large sample scenario.** In Theorems 4.3 and 4.6, we consider the case where the sample size $n$ is less than the input dimension $d$, so that positive definiteness of $XX^\mathsf{T}$ can be established. However, both results still hold for $n > d$. In fact, the squared error loss $\mathcal{L}$ can be written as

$$\sum_{i=1}^{n}\left(f(x_i) - y\right)^2 = \left\|\frac{1}{\sqrt{p}}XW^\mathsf{T}\beta - y\right\|^2 = \left\|\frac{1}{\sqrt{p}}XW^\mathsf{T}\beta - \bar{y}\right\|^2 + \|\bar{y} - y\|^2,$$

where $\bar{y}$ denotes the projection of $y$ onto the column space of $X$. Without loss of generality, we assume $y = \bar{y}$. As a result, $y$ and the columns of $X$ are all in the same $d$-dimensional subspace of $\mathbb{R}^n$ and $XX^\mathsf{T}$ is positive definite on this subspace, as long as $X$ has full column rank. Consequently, we can either work on this subspace of $\mathbb{R}^n$ or project all the vectors onto $\mathbb{R}^d$, and the isometric condition is revised to only consider the $d$ nonzero eigenvalues of $XX^\mathsf{T}$.

## 5 Simulations

Our experiments apply the feedback alignment algorithm to two-layer networks, using a range of networks with different widths and activations. The numerical results suggest that regularization is essential in achieving alignment, in both regression and classification tasks, for linear and nonlinear models. We implement the feedback alignment procedure in PyTorch as an extension of the autograd module for backpropagation, and the training is done on V100 GPUs from internal clusters.

**Feedback alignment on synthetic data.** We first train two-layer networks on synthetic data, where each network $f$ shares the architecture shown in (2.1) and the data are generated by another network $f_0$ that has the same architecture but with random Gaussian weights. We present the experiments for both linear and nonlinear networks, where the activation functions are chosen to be Rectified Linear Unit (ReLU) and hyperbolic tangent (Tanh) for nonlinear case. We set training sample sample size to $n = 50$ and the input dimension $d = 150$, but vary the hidden layer width $p = 100 \times 2^k$ with $k \in [7]$. During training, we take step size $\eta = 10^{-4}$ for linear networks and $\eta = 10^{-3}, 10^{-2}$ for ReLU and Tanh networks, respectively.

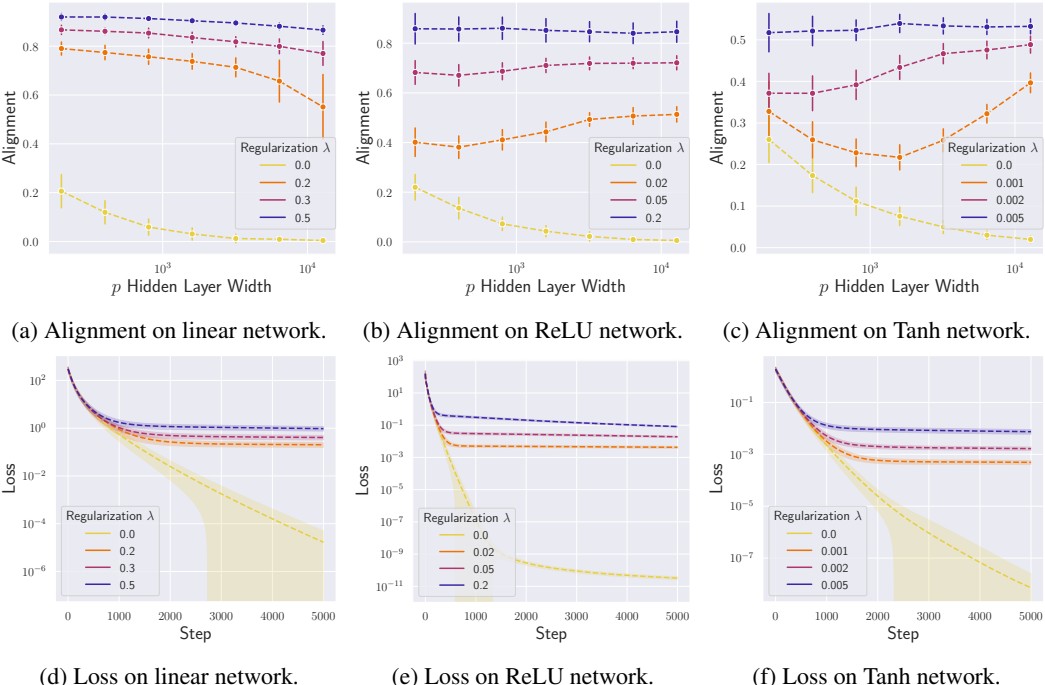

(a) Alignment on linear network.  (b) Alignment on ReLU network.  (c) Alignment on Tanh network.

(d) Loss on linear network.  (e) Loss on ReLU network.  (f) Loss on Tanh network.

Figure 2: Comparisons of alignment and convergence for the feedback alignment algorithm with different levels of $\ell_2$ regularization. In Figs. 2a to 2c, the data points represent the mean value computed across simulations, and the error bars mark the standard deviation out of 50 independent runs. In Figs. 2d to 2f, we show the trajectories of the training loss for networks with $p = 3200$, with the shaded areas indicating the standard deviation over 50 independent runs. The $x$-axes on the first row and the $y$-axes on the second row are presented using a logarithmic scale.

In Figs. 2a to 2c, we show how alignment depends on regularization and the degree of overparameterization as measured by the hidden layer width $p$. Alignment is measured by the cosine of the angle between the forward weights $\beta$ and backward weights $b$. We train the networks until the loss function converges; this procedure is repeated 50 times for each $p$ and $\lambda$. For all three types of networks, as $p$ increases, alignment vanishes if there is no regularization, and grows with the level of regularization $\lambda$ for the same network. We complement the alignment plots with the corresponding loss curves, where the training loss converges slower with larger regularization. These numerical results are consistent with our theoretical statements. Due to the regularization, the loss converges to a positive number that is of the same order as $\lambda$.

We remark that using dropout as a form of regularization can also help the alignment between forward and backward weights (Wager et al., 2013). However, our numerical results suggest that dropout regularization fails to keep the alignment away from zero for networks with large hidden layer width. No theoretical result is available that explains the underlying mechanism.

**Feedback alignment on the MNIST dataset.** The `MNIST` dataset is available under the Creative Commons Attribution-Share Alike 3.0 license (Deng, 2012). It consists of 60,000 training images and 10,000 test images of dimension 28 by 28. We reshape them into vectors of length $d = 784$ and

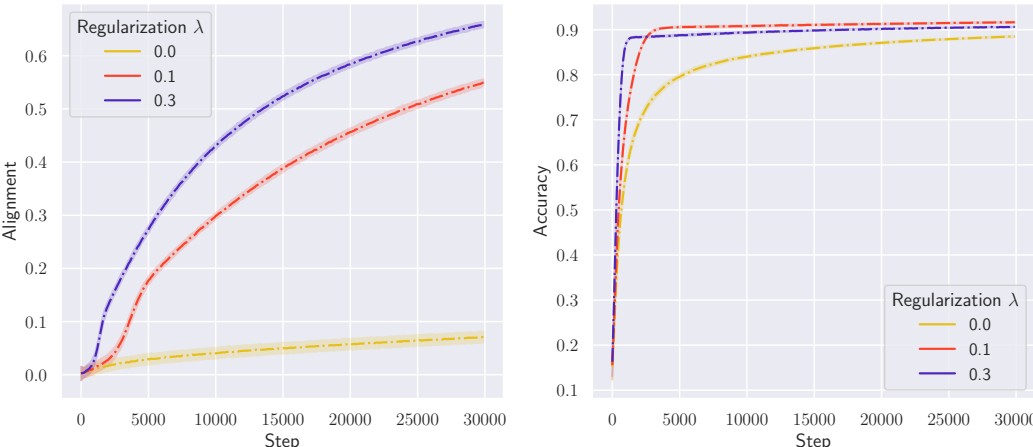

Figure 3: Comparisons on alignment and accuracy for feedback alignment algorithm with $\lambda = 0, 0.1, 0.3$. The left figure shows alignment defined by $\cos\angle(\delta_{\mathrm{BP}}(h), \delta_{\mathrm{FA}}(h))$, and right figure shows the accuracy on the test set. The dashed lines and corresponding shaded areas represent the means and the standard deviations over 10 runs with random initialization.

normalize them by their mean and standard deviation. The network structure is 784-1000-10 with ReLU activation at the hidden layer and with softmax normalization at output layer. During training, we choose the batch size to be 600 and the step size $\eta = 10^{-2}$. The training procedure uses 300 epochs in total. We repeat the training 10 times for each choice of $\lambda$.

Fig. 3 shows the performance of feedback alignment with regularization $\lambda = 0, 0.1, 0.3$. Since the output of the network is not one-dimensional but 10-dimensional, the alignment is now measured by $\cos\angle(\delta_{\mathrm{BP}}(h), \delta_{\mathrm{FA}}(h))$, where $\delta_{\mathrm{BP}}(h)$ is the error signal propagated to the hidden neurons $h$ through forward weights $\beta$, and $\delta_{\mathrm{FA}}(h)$ the error weighted by the random backward weights $b$. We observe that both alignment and convergence are improved by adding regularization to the training, and increasing the regularization level $\lambda$ can further facilitate alignment, with a small gain in test accuracy.

## 6 Discussion

In this paper we have analyzed the feedback alignment algorithm of Lillicrap et al. (2016), showing convergence of the algorithm. The convergence is subtle, as the algorithm does not directly minimize the target loss function; rather, the error is transferred to the hidden neurons through random weights that do not change during the course of learning. The supplement to Lillicrap et al. (2016) presents interesting insights on the dynamics of the algorithm, such as how the feedback weights act as pseudoinverse of the forward weights. After giving an analysis of convergence in the linear case, the authors state that "a general proof must be radically different from those used to demonstrate convergence for backprop" (Supplementary note 16), observing that the algorithm does not minimize any loss function. Our proof of convergence in the general nonlinear case leverages techniques from the use of neural tangent kernel analysis in the over-parameterized setting, but requires more care because the kernel is not positive semi-definite at initialization. In particular, as a sum of two terms $G$ and $H$, the matrix $G$ is concentrated around its postive-definite mean, while $H$ is not generally postive-semidefinite. However, we show that the entries of both matrices remain close to their initial values, due to over-parameterization, and analyze the error term in a Taylor expansion, which establishes convergence.

In analyzing alignment, we unexpectedly found that regularization is essential; without it, the alignment may not persist as the network becomes wider, as our simulations clearly show. Our analysis in the linear case proceeds by essentially showing that

$$\beta(t) = (1 - \eta\lambda)^{t-1}\beta(0) + \frac{\eta}{\sqrt{p}}W(0)X^{\mathsf{T}}\alpha_1(t-1) + \left(\frac{\eta}{\sqrt{p}}\right)b\alpha_2(t-1)$$

and controlling $\alpha_1$ while showing that $\alpha_2$ remains sufficiently large; the regularization kills off the first term. Although we see no obstacle, in principle, to carrying out this proof strategy in the nonlinear case, the calculations are more complex. While convergence requires analysis of the norm of the error, alignment requires understanding the direction of the error. But our simulations strongly suggest this result will go through.

In terms of future research, a technical direction is to extend our results to multilayer networks. It would be interesting to explore local methods to update the backward weights $b$, rather than fixing them, perhaps using a Hebbian update rule in combination with the forward weights $W$. More generally, it is important to study other biologically plausible learning rules that can be implemented in deep learning frameworks at scale and without loss of performance. The results presented here offer support for this as a fruitful line of research. Biologically plausible computational learning contributes to, and shares societal impact with, a large body of fundamental research that aims to understand the basis for cognition in animals, including humans.

## Acknowledgments

Research supported in part by NSF grant CCF-1839308.

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
