# A Convergence on Two-Layer Nonlinear Networks

We consider the family of neural networks

$$f(x) = \frac{1}{\sqrt{p}} \sum_{r=1}^{p} \beta_r \psi(w_r^{\mathsf{T}} x) = \frac{1}{\sqrt{p}} \beta^{\mathsf{T}} \psi(Wx) \tag{A.1}$$

where $\beta \in \mathbb{R}^p$, $W = (w_1, ..., w_p)^{\mathsf{T}} \in \mathbb{R}^{p \times d}$, and $\psi$ is an activation function. Given data, the loss function is

$$\mathcal{L}(W, \beta) = \frac{1}{2} \sum_{i=1}^{n} (f(x_i) - y_i)^2 = \frac{1}{2} \sum_{i=1}^{n} \left( \frac{1}{\sqrt{p}} \beta^{\mathsf{T}} \psi(Wx_i) - y \right)^2. \tag{A.2}$$

The feedback alignment algorithm has updates

$$W(t+1) = W(t) - \eta \frac{1}{\sqrt{p}} \sum_{i=1}^{n} D_i(t) b x_i^{\mathsf{T}} e_i(t)$$

$$\beta(t+1) = \beta(t) - \eta \frac{1}{\sqrt{p}} \sum_{i=1}^{n} \psi(W(t)x_i) e_i(t) \tag{A.3}$$

where $D_i(t) = \operatorname{diag}(\psi'(W(t)x_i))$ and $e_i(t) = \frac{1}{\sqrt{p}} \beta(t)^{\mathsf{T}} \psi(W(t)x_i) - y_i$. To help make the proof more readable, we use $c, C$ to denote the global constants whose values may vary from line to line.

## A.1 Concentration Results

**Lemma A.1** (Lemma A.7 in Gao & Lafferty, 2020). *Assume $x_1, ..., x_n \overset{i.i.d.}{\sim} \mathcal{N}(0, I_d/d)$. We define matrix $\widetilde{G} \in \mathbb{R}^{n \times n}$ with entries*

$$\widetilde{G}_{i,j} = |\mathbb{E}\psi'(Z)|^2 \frac{x_i^{\mathsf{T}} x_j}{\|x_i\| \|x_j\|} + (\mathbb{E}|\psi(Z)|^2 - |\mathbb{E}\psi'(Z)|^2)\mathbb{I}\{i = j\}$$

*where $Z \sim \mathcal{N}(0, 1)$. If $d = \Omega(\log n)$, then with high probability, we have*

$$\|\overline{G} - \widetilde{G}\|^2 \lesssim \frac{\log n}{d} + \frac{n^2}{d^2}.$$

*Proof of Proposition 3.3.* If $\psi$ is sigmoid or tanh, for a standard Gaussian random variable $Z$, we have

$$\gamma := \frac{1}{2}(\mathbb{E}|\psi(Z)|^2 - |\mathbb{E}\psi'(Z)|^2) > 0.$$

From Lemma A.1, we know that with high probability $\lambda_{\min}(\overline{G}) \geq \lambda_{\min}(\widetilde{G}) - \|\overline{G} - \widetilde{G}\| \geq 2\gamma - C(\sqrt{\frac{\log n}{d}} + \frac{n}{d}) \geq \gamma$. $\qquad \square$

**Lemma A.2.** *Assume $W(0)$, $\beta(0)$ and $b$ have i.i.d. standard Gaussian entries. Given $\delta \in (0, 1)$, if $p = \Omega(n/\delta)$, then with probability $1 - \delta$*

$$\frac{1}{p} \sum_{r=1}^{p} |b_r| \leq c, \tag{A.4}$$

$$\frac{1}{p} \sum_{r=1}^{p} |b_r \beta_r(0)| \leq c, \tag{A.5}$$

$$\|e(0)\| \leq c\sqrt{n}, \tag{A.6}$$

$$\max_{r \in [p]} |b_r| \leq 2\sqrt{\log p}. \tag{A.7}$$

*Proof.* We will show each inequality holds with probability at least $1 - \frac{\delta}{4}$, then by a union bound, all of them hold with probability at least $1 - \delta$. Since $\mathbb{Var}(\frac{1}{p} \sum_{r=1}^{p} |b_r|) \leq \frac{\mathbb{Var}(|b_0|)}{p}$, by Chebyshev's inequality, we have

$$\mathbb{P}(\frac{1}{p} \sum_{r=1}^{p} |b_r| > \mathbb{E}(b_1) + 1) \leq \frac{\mathbb{Var}(|b_1|)}{p} \leq \delta/4$$

if $p \geq 4\mathbb{Var}(|b_1|)/\delta$, which gives (A.4). The proof for (A.5) is similar since $\mathbb{Var}(\frac{1}{p} \sum_{r=1}^{p} |b_r \beta_r(0)|) = O(1/p)$. To prove (A.6), since $|y_i|$ and $\|x_i\|$ are bounded, it suffices to show $|u_i(0)| \leq c$ for all $i \in [n]$. Actually, by independence, we have

$$\mathbb{Var}(u_i(0)) = \mathbb{Var}\Big(\frac{1}{p} \sum_{r=1}^{p} \beta_r(0)\psi(w_r(0)^\intercal x_i)\Big) = \frac{1}{p}\mathbb{Var}\Big(\beta_1(0)\psi(w_1(0)^\intercal x_i)\Big) = O(1/p).$$

By Chebyshev's inequality, we have for each $i \in [n]$

$$\mathbb{P}(|u_i(0)| > c) \leq \frac{\mathbb{Var}(u_i(0))}{c^2} \leq \frac{\delta}{4n}$$

where we require $p = \Omega(n/\delta)$. With a union bound argument, we can show (A.6). Finally, (A.7) followed from standard Gaussian tail bounds and union bound argument, yielding

$$\mathbb{P}(\max_{r \in [p]} |b_r| > 2\sqrt{\log p}) \leq \sum_{r \in [p]} \mathbb{P}(|b_r| > 2\sqrt{\log p}) \leq 2pe^{-2\log p} = \frac{2}{p} \leq \frac{\delta}{4}.$$

$\square$

**Lemma A.3.** *Under the conditions of Theorem 3.2, we define matrices $G(0), H(0) \in \mathbb{R}^{n \times n}$ with entries*

$$G_{ij}(0) = \frac{1}{p}\psi(W(0)x_i)^\intercal \psi(W(0)x_j) = \frac{1}{p}\sum_{r=1}^{p} \psi(w_r(0)^\intercal x_i)\psi(w_r(0)^\intercal x_j) \qquad \text{(A.8)}$$

*and*

$$H_{ij}(0) = \frac{x_i^\intercal x_j}{p}\beta(0)^\intercal D_i(0)D_j(0)b = \frac{1}{p}\sum_{r=1}^{p} \beta_r(0)b_r\psi'(w_r(0)^\intercal x_i)\psi'(w_r(0)^\intercal x_j). \qquad \text{(A.9)}$$

*For any $\delta \in (0,1)$, if $p = \Omega(\frac{n^2}{\delta\gamma^2})$, then with probability at least $1 - \delta$, we have $\lambda_{\min}(G(0)) \geq \frac{3}{4}\gamma$ and $\|H(0)\| \leq \frac{\gamma}{4}$.*

*Proof.* By independence and boundedness of $\psi$ and $\psi'$, we have $\mathbb{Var}(G_{ij}(0)) = O(1/p)$ and $\mathbb{Var}(H_{ij}(0)) = O(1/p)$. Since $\mathbb{E}(G(0)) = \overline{G}$, we have

$$\mathbb{E}\|G(0) - \overline{G}\|^2 \leq \mathbb{E}\|G(0) - \overline{G}\|_F^2 = O(\frac{n^2}{p}).$$

By Markov's inequality, when $p = \Omega(\frac{n^2}{\delta\gamma^2})$

$$\mathbb{P}(\|G(0) - \overline{G}\| > \frac{\gamma}{4}) \leq O(\frac{n^2}{p\gamma^2}) \leq \frac{\delta}{2}.$$

Similarly we have $\mathbb{P}(\|H(0)\| > \frac{\gamma}{4}) \leq \frac{\delta}{2}$, since $\mathbb{E}(H(0)) = 0$. Then with probability at least $1 - \delta$, $\lambda_{\min}(G(0)) \geq \lambda_{\min}(\overline{G}) - \gamma/4 \geq \frac{3}{4}\gamma$, and $\|H(0)\| \leq \gamma/4$. $\square$

## A.2  Proof of Theorem 3.2

**Lemma A.4.** *Assume all the inequalities from Lemma A.2 hold. Under the conditions of Theorem 3.2, if the error bound (3.1) holds for all $t = 1, 2, ..., t' - 1$, then the bounds (3.2) hold for all $t \leq t'$.*

*Proof.* From the feedback alignment updates (A.3), we have for all $t \leq T$

$$|\beta_r(t) - \beta_r(0)| \leq \frac{\eta}{\sqrt{p}} \sum_{s=0}^{t-1} \sum_{i=1}^{n} |\psi(w_r(t)x_i)e_i(t)|$$

$$\leq c \frac{\eta}{\sqrt{p}} \sum_{s=0}^{t-1} \sum_{i=1}^{n} |e_i(t)|$$

$$\leq c \frac{\eta\sqrt{n}}{\sqrt{p}} \sum_{s=0}^{t-1} \|e(t)\|$$

$$\leq c \frac{\eta\sqrt{n}}{\sqrt{p}} \sum_{s=0}^{t-1} (1 - \frac{\gamma\eta}{4})^t \|e(0)\|$$

$$\leq c \frac{\sqrt{n}}{\gamma\sqrt{p}} \|e(0)\|$$

$$\leq c \frac{n}{\gamma\sqrt{p}}$$

where we use the fact that $\psi$ is bounded and (A.6). We also have

$$\|w_r(t) - w_r(0)\| \leq \frac{\eta}{\sqrt{p}} \sum_{s=0}^{t-1} \sum_{i=1}^{n} \|\psi'(w_r(t)^\mathsf{T} x_i)b_r x_i e_i(t)\|$$

$$\leq c \frac{\eta}{\sqrt{p}} \sum_{s=0}^{t-1} \sum_{i=1}^{n} |b_r| |e_i(t)|$$

$$\leq c |b_r| \frac{\eta\sqrt{n}}{\sqrt{p}} \sum_{s=0}^{t-1} \|e(t)\|$$

$$\leq c |b_r| \frac{\sqrt{n}}{\gamma\sqrt{p}} \|e(0)\|$$

$$\leq c \frac{n\sqrt{\log p}}{\gamma\sqrt{p}}$$

where we use that $\psi'$ is bounded, (A.6) and (A.7). $\qquad\square$

**Lemma A.5.** *Assume all the inequalities from Lemma A.2 hold. Under the conditions of Theorem 3.2, if the bound for the weights difference (3.2) holds for all $t \leq t'$ and error bound (3.1) holds for all $t \leq t' - 1$, then (3.1) holds for $t = t'$.*

*Proof.* We start with analyzing the error $e(t)$ according to

$$e_i(t+1) = \frac{1}{\sqrt{p}} \beta(t+1)^\mathsf{T} \psi(W(t+1)x_i) - y_i$$

$$= \frac{1}{\sqrt{p}} \beta(t+1)^\mathsf{T} (\psi(W(t+1)x_i) - \psi(W(t)x_i)) + \frac{1}{\sqrt{p}} (\beta(t+1) - \beta(t))^\mathsf{T} \psi(W(t)x_i)$$

$$\quad + \frac{1}{\sqrt{p}} \beta(t)^\mathsf{T} \psi(W(t)x_i) - y_i$$

$$= e_i(t) - \frac{\eta}{p} \beta(t+1)^\mathsf{T} D_i(t) \sum_{j=1}^{n} D_j(t) b x_j^\mathsf{T} x_i e_j(t) - \frac{\eta}{p} \sum_{j=1}^{n} \psi(W(t)x_j)^\mathsf{T} \psi(W(t)x_i) e_j(t)$$

$$\quad + v_i(t)$$

$$= e_i(t) - \eta \sum_{j=1}^{n} (H_{ij}(t) + G_{ij}(t)) e_j(t) + v_i(t)$$

where

$$G_{ij}(t) = \frac{1}{p}\psi(W(t)x_j)^{\mathsf{T}}\psi(W(t)x_i)$$

$$H_{ij}(t) = \frac{x_i^{\mathsf{T}}x_j}{p}\beta(t+1)^{\mathsf{T}}D_i(t)D_j(t)b$$

and $v_i(t)$ is the residual term from the Taylor expansion

$$v_i(t) = \frac{1}{2\sqrt{p}}\sum_{r=1}^{p}\beta_r(t+1)|(w_r(t+1)-w_r(t))^{\mathsf{T}}x_i|^2\psi''(\xi_{ri}(t))$$

with $\xi_{ri}(t)$ between $w_r(t)^{\mathsf{T}}x_i$ and $w_r(t+1)^{\mathsf{T}}x_i$. We can also rewrite the above iteration in vector form as

$$e(t+1) = e(t) - \eta(G(t)+H(t))e(t) + v(t). \tag{A.10}$$

Now for $t = t'-1$, we wish to show that both $G(t)$ and $H(t)$ are close to their initialization. Notice that

$$
\begin{aligned}
|G_{ij}(t) - G_{ij}(0)| &= \frac{1}{p}\left|\psi(W(t)x_j)^{\mathsf{T}}\psi(W(t)x_i) - \psi(W(t)x_j)^{\mathsf{T}}\psi(W(t)x_i)\right| \\
&\leq \frac{1}{p}\sum_{r=1}^{p}|\psi(w_r(t)^{\mathsf{T}}x_j)||\psi(w_r(t)^{\mathsf{T}}x_i) - \psi(w_r(0)^{\mathsf{T}}x_i)| \\
&\quad + \frac{1}{p}\sum_{r=1}^{p}|\psi(w_r(0)^{\mathsf{T}}x_i)||\psi(w_r(t)^{\mathsf{T}}x_j) - \psi(w_r(0)^{\mathsf{T}}x_j)| \\
&\leq c\frac{1}{p}\sum_{r=1}^{p}|w_r(t)^{\mathsf{T}}x_i - w_r(0)^{\mathsf{T}}x_i| + \frac{1}{p}\sum_{r=1}^{p}|w_r(t)^{\mathsf{T}}x_j - w_r(0)^{\mathsf{T}}x_j| \\
&\leq c_0\frac{n\sqrt{\log p}}{\gamma\sqrt{p}}(\|x_i\| + \|x_j\|)
\end{aligned}
$$

where the second inequality is due to the boundedness of $\psi$ and $\psi'$, and the last inequality is by (3.2). Then we have

$$\|G(t) - G(0)\| \leq \max_{j\in[n]}\sum_{i=1}^{n}|G_{ij}(t) - G_{ij}(0)| \leq c_0\frac{n^2\sqrt{\log p}}{\gamma\sqrt{p}}. \tag{A.11}$$

For matrix $H(t)$, we similarly have

$$
\begin{aligned}
|H_{ij}(t) - H_{ij}(0)| &\leq \frac{|x_i^{\mathsf{T}}x_j|}{p}\left|\beta(t+1)^{\mathsf{T}}D_i(t)D_j(t)b - \beta(0)^{\mathsf{T}}D_i(0)D_j(0)b\right| \\
&\leq \frac{\|x_i\|\|x_j\|}{p}\sum_{r=1}^{p}\Big|b_r\beta_r(t+1)\psi'(w_r(t)^{\mathsf{T}}x_i)\psi'(w_r(t)^{\mathsf{T}}x_j) \\
&\quad - b_r\beta_r(0)\psi'(w_r(0)^{\mathsf{T}}x_i)\psi'(w_r(0)^{\mathsf{T}}x_j)\Big| \\
&\leq \frac{\|\|x_i\|\|x_j\|\|}{p}\sum_{r=1}^{p}\Big(|b_r||\beta_r(t+1)-\beta_r(0)||\psi'(w_r(t)^{\mathsf{T}}x_i)\psi'(w_r(t)^{\mathsf{T}}x_j)| \\
&\quad + |b_r||\beta_r(0)||\psi'(w_r(t)^{\mathsf{T}}x_i) - \psi'(w_r(0)^{\mathsf{T}}x_i)||\psi'(w_r(t)^{\mathsf{T}}x_j)| \\
&\quad + |b_r||\beta_r(0)||\psi'(w_r(0)^{\mathsf{T}}x_i)||\psi'(w_r(t)^{\mathsf{T}}x_j) - \psi'(w_r(0)^{\mathsf{T}}x_j)|\Big) \\
&\leq c\frac{\|x_i\|\|x_j\|}{p}\sum_{r=1}^{p}\Big(|b_r|\frac{n}{\gamma\sqrt{p}} + |b_r||\beta_r(0)|\frac{n\sqrt{\log p}}{\gamma\sqrt{p}}(\|x_i\|+\|x_j\|)\Big) \\
&\leq c_1\frac{n}{\gamma\sqrt{p}} + c_2\frac{n\sqrt{\log p}}{\gamma\sqrt{p}}.
\end{aligned}
$$

It follows that

$$\|H(t) - H(0)\| \le \max_{j \in [n]} \sum_{i=1}^{n} |H_{ij}(t) - H_{ij}(0)| \le c_1 \frac{n^2}{\gamma\sqrt{p}} + c_2 \frac{n^2\sqrt{\log p}}{\gamma\sqrt{p}}. \tag{A.12}$$

Next, we bound the residual term $v_i(t)$. Since $\psi''$ is bounded, we have

$$
\begin{aligned}
|v_i(t)| &\le c\frac{1}{\sqrt{p}} \sum_{r=1}^{p} |\beta_r(t+1)| \|w_r(t+1) - w_r(t)\|^2 \\
&\le c\frac{1}{\sqrt{p}} \frac{\eta^2}{p} \sum_{r=1}^{p} |\beta_r(t+1)| \Big( \sum_{i=1}^{n} \|\psi'(w_r(t)^\mathsf{T} x_i) b_r x_i e_i(t)\| \Big)^2 \\
&\le c\frac{1}{\sqrt{p}} \frac{\eta^2}{p} \sum_{r=1}^{p} |\beta_r(t+1)| |b_r|^2 \Big( \sum_{i=1}^{n} |e_i(t)| \Big)^2 \\
&\le c\frac{\eta^2 n}{\sqrt{p}} \|e(t)\|^2 \\
&\le c_3 \frac{\eta^2 n\sqrt{n}}{\sqrt{p}} \|e(t)\|.
\end{aligned}
$$

This leads to the bound

$$\|v(t)\| = \Big( \sum_{i=1}^{n} |v_i(t)|^2 \Big)^{1/2} \le c_3 \frac{\eta^2 n^2}{\sqrt{p}} \|e(t)\|. \tag{A.13}$$

Combining Eqs. (A.10) to (A.13), we have

$$
\begin{aligned}
\|e(t+1)\| &\le \|I_n - \eta(G(t) + H(t))\| \|e(t)\| + \|v(t)\| \\
&\le \Big( \|I_n - \eta G(0)\| + \eta\|G(t) - G(0)\| + \eta\|H(0)\| \\
&\quad + \eta\|H(t) - H(0)\| \Big) \|e(t)\| + \|v(t)\| \\
&\le \Big( 1 - \frac{3\eta\gamma}{4} + c_0 \frac{\eta n^2\sqrt{\log p}}{\gamma\sqrt{p}} + \frac{\eta\gamma}{4} + c_1 \frac{\eta n^2}{\gamma\sqrt{p}} + c_2 \frac{\eta n^2\sqrt{\log p}}{\gamma\sqrt{p}} + c_3 \frac{\eta^2 n\sqrt{n}}{\sqrt{p}} \Big) \|e(t)\| \\
&\le (1 - \frac{\eta\gamma}{4}) \|e(t)\|
\end{aligned}
$$

where we use Lemma A.3 and $p = \Omega(\frac{n^4 \log p}{\gamma^4})$. $\qquad\square$

*Proof of Theorem 3.2.* We prove the inequality (3.1) by induction. Suppose (3.1) and (3.2) hold for all $t = 1, 2, ..., t' - 1$, by Lemma A.4 and Lemma A.5 we know (3.1) and (3.2) hold for $t = t'$, which completes the proof. $\qquad\square$

### A.3 Proof of Theorem 4.2

**Lemma A.6.** *Assume all the inequalities from Lemma A.2 hold. Under the conditions of Theorem 4.2, if the error bound (4.2) holds for all $t = 1, 2, ..., t' - 1$, then*

$$
\begin{aligned}
\|w_r(t) - w_r(0)\| &\le c_1 \frac{n\sqrt{\log p}}{\gamma\sqrt{p}} (1 + \eta\tilde{S}_\lambda), \\
|\beta_r(t) - \beta_r(0)| &\le c_2 \frac{n}{\gamma\sqrt{p}} (1 + \eta\tilde{S}_\lambda)
\end{aligned} \tag{A.14}
$$

*hold for all $t \le t'$, where $c_1$, $c_2$ are constants.*

*Proof.* For any $k \le t' - 1$, we apply (4.2) repeatedly on the right hand side of itself to get

$$\|e(k)\| \le \prod_{i=0}^{k-1} \Big( 1 - \frac{\eta\gamma}{4} - \eta\lambda(i) \Big) \|e(0)\| + \sum_{i=0}^{k-1} \eta\lambda(i) \prod_{i<j<k} \Big( 1 - \frac{\eta\gamma}{4} - \eta\lambda(j) \Big) \|y\|.$$

For $t \leq t' - 1$, we take the sum over $k = 0, .., t$ on both sides of above inequality to obtain

$$\sum_{k=0}^{t} \|e(k)\| \leq \sum_{k=0}^{t} \prod_{i=0}^{k-1} \left(1 - \frac{\eta\gamma}{4} - \eta\lambda(i)\right) \|e(0)\| + \sum_{k=0}^{t} \sum_{i=0}^{k-1} \eta\lambda(i) \prod_{i<j<k} \left(1 - \frac{\eta\gamma}{4} - \eta\lambda(j)\right) \|y\|$$

$$\leq \sum_{k=0}^{t} \left(1 - \frac{\eta\gamma}{4}\right)^{k-1} \|e(0)\| + \sum_{k=0}^{t} \sum_{i=0}^{k-1} \eta\lambda(i) \left(1 - \frac{\eta\gamma}{4}\right)^{k-i-1} \|y\|$$

$$\leq \sum_{k=0}^{t} \left(1 - \frac{\eta\gamma}{4}\right)^{k-1} \|e(0)\| + \eta\|y\| \sum_{k=0}^{t-1} \lambda(i) \sum_{k=i+1}^{T} \left(1 - \frac{\eta\gamma}{4}\right)^{k-i-1}$$

$$\leq \frac{4}{\eta\gamma} \|e(0)\| + \frac{4}{\gamma} \tilde{S}_\lambda \|y\|$$

$$\leq \frac{c\sqrt{n}}{\gamma} \left(\frac{1}{\eta} + \tilde{S}_\lambda\right)$$

where we use $\|e(0)\| = O(\sqrt{n})$ and $\|y\| = O(\sqrt{n})$. Then for all $t \leq t'$, we have

$$|\beta_r(t) - \beta_r(0)| \leq \frac{\eta}{\sqrt{p}} \sum_{s=0}^{t-1} \sum_{i=1}^{n} |\psi(w_r(t)x_i)e_i(t)|$$

$$\leq c\frac{\eta}{\sqrt{p}} \sum_{s=0}^{t-1} \sum_{i=1}^{n} |e_i(t)|$$

$$\leq c\frac{\eta\sqrt{n}}{\sqrt{p}} \sum_{s=0}^{t-1} \|e(t)\|$$

$$\leq c\frac{\eta\sqrt{n}}{\sqrt{p}} \frac{\sqrt{n}}{\gamma} \left(\frac{1}{\eta} + \tilde{S}_\lambda\right)$$

$$\leq c\frac{n}{\gamma\sqrt{p}} (1 + \eta\tilde{S}_\lambda)$$

where we use $\psi$ is bounded and (A.6). We also have

$$\|w_r(t) - w_r(0)\| \leq \frac{\eta}{\sqrt{p}} \sum_{s=0}^{t-1} \sum_{i=1}^{n} \|\psi'(w_r(t)^\mathsf{T} x_i)b_r x_i e_i(t)\|$$

$$\leq c\frac{\eta}{\sqrt{p}} \sum_{s=0}^{t-1} \sum_{i=1}^{n} |b_r||e_i(t)|$$

$$\leq c|b_r|\frac{\eta\sqrt{n}}{\sqrt{p}} \sum_{s=0}^{t-1} \|e(t)\|$$

$$\leq c|b_r|\frac{\eta\sqrt{n}}{\sqrt{p}} \frac{\sqrt{n}}{\gamma} \left(\frac{1}{\eta} + \tilde{S}_\lambda\right)$$

$$\leq c\frac{n\sqrt{\log p}}{\gamma\sqrt{p}} (1 + \eta\tilde{S}_\lambda)$$

where we use the fact that $\psi'$ is bounded, (A.6) and (A.7). □

**Lemma A.7.** *Assume all the inequalities from Lemma A.2 hold. Under the conditions of Theorem 4.2, if the bound for weights difference (A.14) holds for all $t \leq t'$ and error bound (4.2) holds for all $t \leq t' - 1$, then (4.2) holds for $t = t'$.*

*Proof.* We start by analyzing the error $e(t)$ according to

$$
\begin{aligned}
e_i(t+1) &= \frac{1}{\sqrt{p}}\beta(t+1)^{\mathsf{T}}\psi(W(t+1)x_i) - y_i \\
&= \frac{1}{\sqrt{p}}\beta(t+1)^{\mathsf{T}}(\psi(W(t+1)x_i) - \psi(W(t)x_i)) + \frac{1}{\sqrt{p}}(\beta(t+1) - (1-\eta\lambda(t))\beta(t))^{\mathsf{T}}\psi(W(t)x_i) \\
&\quad + (1-\eta\lambda(t))\Big(\frac{1}{\sqrt{p}}\beta(t)^{\mathsf{T}}\psi(W(t)x_i) - y_i\Big) - \eta\lambda(t)y \\
&= (1-\eta\lambda(t))e_i(t) - \frac{\eta}{p}\beta(t+1)^{\mathsf{T}}D_i(t)\sum_{j=1}^{n}D_j(t)bx_j^{\mathsf{T}}x_i e_j(t) - \frac{\eta}{p}\sum_{j=1}^{n}\psi(W(t)x_j)^{\mathsf{T}}\psi(W(t)x_i)e_j(t) - \eta\lambda(t)y \\
&\quad + v_i(t) \\
&= (1-\eta\lambda(t))e_i(t) - \eta\sum_{j=1}^{n}\big(H_{ij}(t) + G_{ij}(t)\big)e_j(t) + v_i(t) - \eta\lambda(t)y
\end{aligned}
$$

where

$$
G_{ij}(t) = \frac{1}{p}\psi(W(t)x_j)^{\mathsf{T}}\psi(W(t)x_i)
$$

$$
H_{ij}(t) = \frac{x_i^{\mathsf{T}}x_j}{p}\beta(t+1)^{\mathsf{T}}D_i(t)D_j(t)b
$$

and $v_i(t)$ is the residual term from a Taylor expansion

$$
v_i(t) = \frac{1}{2\sqrt{p}}\sum_{r=1}^{p}\beta_r(t+1)|(w_r(t+1) - w_r(t))^{\mathsf{T}}x_i|^2\psi''(\xi_{ri}(t))
$$

with $\xi_{ri}(t)$ between $w_r(t)^{\mathsf{T}}x_i$ and $w_r(t+1)^{\mathsf{T}}x_i$. We can also rewrite the above iteration in vector form as

$$
e(t+1) = (1-\lambda(t))e(t) - \eta(G(t) + H(t))e(t) + v(t) - \eta\lambda(t)y. \tag{A.15}
$$

Now for $t = t'-1$, we show that both $G(t)$ and $H(t)$ are close to their initialization. Using the argument in Lemma A.5, we can obtain following bounds

$$
\|G(t) - G(0)\| \leq c_1 \frac{n^2\sqrt{\log p}}{\gamma\sqrt{p}}(1+\eta\tilde{S}_\lambda) \tag{A.16}
$$

$$
\|H(t) - H(0)\| \leq c_2 \frac{n^2\sqrt{\log p}}{\gamma\sqrt{p}}(1+\eta\tilde{S}_\lambda) \tag{A.17}
$$

$$
\|v(t)\| \leq c_3 \frac{\eta^2 n^2}{\sqrt{p}}\|e(t)\|. \tag{A.18}
$$

Combining Eqs. (A.15) to (A.18), we have

$$
\begin{aligned}
\|e(t+1)\| &\leq \|(1-\eta\lambda(t))I_n - \eta(G(t) + H(t))\|\|e(t)\| + \|v(t)\| \\
&\leq \Big(\|(1-\eta\lambda(t))I_n - \eta G(0)\| + \eta\|G(t) - G(0)\| + \eta\|H(0)\| \\
&\quad + \eta\|H(t) - H(0)\|\Big)\|e(t)\| + \|v(t)\| \\
&\leq \Big(1 - \eta\lambda(t) - \frac{3\eta\gamma}{4} + (c_1 + c_2)\frac{\eta n^2\sqrt{\log p}}{\gamma\sqrt{p}}(1+\eta\tilde{S}_\lambda) + c_3\frac{\eta^2 n\sqrt{n}}{\sqrt{p}}\Big)\|e(t)\| \\
&\leq (1 - \eta\lambda(t) - \frac{\eta\gamma}{4})\|e(t)\|
\end{aligned}
$$

where we use Lemma A.3, $p = \Omega(\frac{n^4\log p}{\gamma^4})$ and $\tilde{S}_\lambda = O(\frac{\gamma^2\sqrt{p}}{\eta n^2\sqrt{\log p}})$. $\qquad\square$

*Proof of Theorem 4.2.* We prove the inequality (4.2) by induction. Suppose (4.2) holds for all $t = 1, 2, ..., t'-1$. Then by Lemma A.6 and Lemma A.7 we know (4.2) holds for $t = t'$, which completes the proof. $\qquad\square$

# B  Alignment on Two-Layer Linear Networks

Now we assume $\psi(u) = u$, so that $f$ is a linear network. The loss function with regularization at time $t$ is

$$\mathcal{L}(t, W, \beta) = \frac{1}{2}\left\|\frac{1}{\sqrt{p}}XW^\intercal\beta - y\right\|^2 + \frac{1}{2}\lambda(t)\|\beta\|^2. \tag{B.1}$$

The regularized feedback alignment algorithm gives

$$W(t+1) = W(t) - \eta\frac{1}{\sqrt{p}}be(t)^\intercal X$$
$$\beta(t+1) = (1 - \eta\lambda(t))\beta(t) - \frac{\eta}{\sqrt{p}}W(t)X^\intercal e(t) \tag{B.2}$$

where $e(t) = \frac{1}{\sqrt{p}}XW(t)^\intercal\beta(t) - y$ is the error vector at time t.

**Lemma B.1.** *Suppose the network is trained with the regularized feedback alignment algorithm* (B.2). *Then the prediction error $e(t)$ satisfies the recurrence*

$$e(t+1) = \left[(1 - \eta\lambda(t))I_d - \frac{\eta}{p}XW(0)^\intercal W(0)X^\intercal - \eta\Big(J_1(t) + J_2(t) + J_3(t)\Big)\right]e(t) - \eta\lambda(t)y \tag{B.3}$$

*where*

$$J_1(t) = \frac{1}{p}b^\intercal\beta(0)\prod_{i=0}^{t}(1 - \eta\lambda(i))XX^\intercal$$
$$J_2(t) = -\frac{\eta}{p}\Big(\bar{v}^\intercal X^\intercal\hat{s}(t)XX^\intercal + XX^\intercal s(t-1)\bar{v}^\intercal X^\intercal + X\bar{v}s(t-1)^\intercal XX^\intercal\Big)$$
$$J_3(t) = \frac{\eta^2}{p^2}\|b\|^2\Big(\hat{S}(t)XX^\intercal + XX^\intercal s(t-1)s(t-1)^\intercal XX^\intercal\Big)$$

*and*

$$\bar{v} = \frac{1}{\sqrt{p}}W(0)^\intercal b$$
$$s(t) = \sum_{i=0}^{t}e(i)$$
$$\hat{s}(t) = \sum_{i=0}^{t}\prod_{i<k\leq t}(1 - \eta\lambda(k))e(i)$$
$$\hat{S}(t) = \sum_{i=0}^{t}\prod_{i<k\leq t}(1 - \eta\lambda(k))e(i)^\intercal XX^\intercal\sum_{j=0}^{i-1}e(j).$$

*Proof.* We first write $W(t)$ in terms of $W(0)$ and $e(i)$, $i \in [t]$, so that

$$W(t) = W(0) - \frac{\eta}{\sqrt{p}}b\sum_{i=0}^{t-1}e(i)^\intercal X = W(0) - \frac{\eta}{\sqrt{p}}bs(t-1)^\intercal X. \tag{B.4}$$

Similarly, for $\beta(t)$ we have

$$
\begin{aligned}
\beta(t) &= \prod_{i=0}^{t-1}(1-\eta\lambda(i))\beta(0) - \frac{\eta}{\sqrt{p}}\sum_{i=0}^{t-1}\prod_{i<k<t}(1-\eta\lambda(k))W(i)X^\mathsf{T}e(i) \\
&= \prod_{i=0}^{t-1}(1-\eta\lambda(i))\beta(0) - \frac{\eta}{\sqrt{p}}\sum_{i=0}^{t-1}\prod_{i<k<t}(1-\eta\lambda(k))\Big(W(0)-\frac{\eta}{\sqrt{p}}b\sum_{j=0}^{i-1}e(j)^\mathsf{T}X\Big)X^\mathsf{T}e(i) \\
&= \prod_{i=0}^{t-1}(1-\eta\lambda(i))\beta(0) - \frac{\eta}{\sqrt{p}}\sum_{i=0}^{t-1}\prod_{i<k<t}(1-\eta\lambda(k))W(0)X^\mathsf{T}e(i) \\
&\quad + \frac{\eta^2}{p}b\sum_{i=0}^{t-1}\prod_{i<k<t}(1-\eta\lambda(k))e(i)^\mathsf{T}XX^\mathsf{T}\sum_{j=0}^{i-1}e(j) \\
&= \prod_{i=0}^{t-1}(1-\eta\lambda(i))\beta(0) - \frac{\eta}{\sqrt{p}}W(0)X^\mathsf{T}\hat{s}(t-1) + \frac{\eta^2}{p}b\hat{S}(t-1).
\end{aligned}
\tag{B.5}
$$

We now study how the error $e(t)$ changes after a single update step, writing

$$
\begin{aligned}
e(t+1) &= \frac{1}{\sqrt{p}}XW(t+1)^\mathsf{T}\beta(t+1) - y \\
&= \frac{1}{\sqrt{p}}X(W(t+1)-W(t)^\mathsf{T}\beta(t+1) + \frac{1}{\sqrt{p}}XW(t)^\mathsf{T}(\beta(t+1)-(1-\eta\lambda(t))\beta(t)) \\
&\quad + (1-\eta\lambda(t))\Big(\frac{1}{\sqrt{p}}XW(t)^\mathsf{T}\beta(t)-y\Big) - \eta\lambda(t)y \\
&= (1-\eta\lambda(t))e(t) - \frac{\eta}{p}b^\mathsf{T}\beta(t+1)XX^\mathsf{T}e(t) - \frac{\eta}{p}XW(t)^\mathsf{T}W(t)X^\mathsf{T}e(t) - \eta\lambda(t)y
\end{aligned}
$$

By plugging (B.4) and (B.5) into above equation, we have

$$
\begin{aligned}
e(t+1) &= (1-\eta\lambda(t))e(t) \\
&\quad - \frac{\eta}{p}b^\mathsf{T}\Big[\prod_{i=0}^{t}(1-\eta\lambda(i))\beta(0) - \frac{\eta}{\sqrt{p}}W(0)X^\mathsf{T}\hat{s}(t) + \frac{\eta^2}{p}b\hat{S}(t)\Big]XX^\mathsf{T}e(t) \\
&\quad - \frac{\eta}{p}X\Big[W(0)-\frac{\eta}{\sqrt{p}}bs(t-1)^\mathsf{T}X\Big]^\mathsf{T}\Big[W(0)-\frac{\eta}{\sqrt{p}}bs(t-1)^\mathsf{T}X\Big]X^\mathsf{T}e(t) \\
&\quad - \eta\lambda(t)y
\end{aligned}
$$

After expanding the brackets and rearranging the items, we can obtain (B.3). $\qquad\square$

**Lemma B.2.** *Given $\delta \in (0,1)$ and $\epsilon > 0$ , if $p = \Omega(\frac{1}{\epsilon}\log\frac{d}{\delta} + \frac{d}{\epsilon}\log\frac{1}{\epsilon})$, the following inequalities hold with probability at least $1-\delta$*

$$
\frac{|b^\mathsf{T}\beta(0)|}{\sqrt{p}} \le c\sqrt{\log\frac{1}{\delta}}
\tag{B.6}
$$

$$
\frac{\|b^\mathsf{T}W(0)\|}{\sqrt{p}} \le c\sqrt{d\log\frac{d}{\delta}}
\tag{B.7}
$$

$$
\Big|\frac{\|b\|^2}{p} - 1\Big| \le \frac{c}{\sqrt{p}}\sqrt{\log\frac{1}{\delta}}
\tag{B.8}
$$

$$
\Big\|\frac{1}{p}W(0)^\mathsf{T}W(0) - I_d\Big\| \le \epsilon
\tag{B.9}
$$

*where $c$ is a constant.*

*Proof.* (B.6) is derived from Lemma C.4. (B.7) is by (B.6) and a union bound argument. (B.8) is by Lemma C.3. (B.9) is by Corollary C.2 $\qquad\square$

*Proof of Theorem 4.3.* We show (4.3) by induction. Assume (4.3) holds for all $t = 0, 1, ..., t'$, we will show it hold for $t = t' + 1$. For any $k \leq t'$, we apply (4.3) repeatedly on the right hand side of itself to get

$$\|e(k)\| \leq \prod_{i=0}^{k-1} \left(1 - \frac{\eta\gamma}{2} - \eta\lambda(i)\right)\|e(0)\| + \sum_{i=0}^{k-1} \eta\lambda(i) \prod_{i<j<k} \left(1 - \frac{\eta\gamma}{2} - \eta\lambda(j)\right)\|y\|$$

For $t \leq t'$, we take the sum over $k = 0, .., t$ on both sides of above inequality

$$\sum_{k=0}^{t} \|e(k)\| \leq \sum_{k=0}^{t} \prod_{i=0}^{k-1} \left(1 - \frac{\eta\gamma}{2} - \eta\lambda(i)\right)\|e(0)\| + \sum_{k=0}^{t} \sum_{i=0}^{k-1} \eta\lambda(i) \prod_{i<j<k} \left(1 - \frac{\eta\gamma}{2} - \eta\lambda(j)\right)\|y\|$$

$$\leq \sum_{k=0}^{t} \left(1 - \frac{\eta\gamma}{2}\right)^{k-1}\|e(0)\| + \sum_{k=0}^{t} \sum_{i=0}^{k-1} \eta\lambda(i)\left(1 - \frac{\eta\gamma}{2}\right)^{k-i-1}\|y\|$$

$$\leq \sum_{k=0}^{t} \left(1 - \frac{\eta\gamma}{2}\right)^{k-1}\|e(0)\| + \eta\|y\| \sum_{k=0}^{t-1} \lambda(i) \sum_{k=i+1}^{T} \left(1 - \frac{\eta\gamma}{2}\right)^{k-i-1}$$

$$\leq \frac{2}{\eta\gamma}\|e(0)\| + \frac{2}{\gamma}S_\lambda\|y\|$$

$$\leq \frac{c\sqrt{n}}{\gamma}\left(\frac{1}{\eta} + S_\lambda\right)$$

where we use $\|e(0)\| = O(\sqrt{n})$ and $\|y\| = O(\sqrt{n})$. With this bound and the inequalities from Lemma B.2, we can bound the norms of $J_1(t)$, $J_2(t)$ and $J_3(t)$ from Lemma B.1. It follows that

$$\|J_1(t)\| \leq \frac{1}{p}|b^{\mathsf{T}}\beta(0)|\|XX^{\mathsf{T}}\| \leq c\frac{M\sqrt{\log \delta^{-1}}}{\sqrt{p}} \leq \frac{\gamma}{16}, \tag{B.10}$$

$$\|J_2(t)\| \leq \frac{\eta}{p}\|X\|\|XX^{\mathsf{T}}\|\|\bar{v}\|(2\|s(t-1)\| + \|\hat{s}(t)\|) \leq c\frac{\eta}{p}M^{3/2}\sqrt{d\log\frac{d}{\delta}}\frac{\sqrt{n}}{\gamma}\left(\frac{1}{\eta} + S_\lambda\right) \leq \frac{\gamma}{16} \tag{B.11}$$

and

$$\|J_3(t)\| \leq \frac{\eta^2}{p^2}\|b\|^2(\|XX^{\mathsf{T}}\|\|\hat{S}(t)\| + \|XX^{\mathsf{T}}\|^2\|s(t-1)\|^2) \leq c\frac{\eta^2}{p}M^2\frac{n}{\gamma^2}\left(\frac{1}{\eta} + S_\lambda\right)^2 \leq \frac{\gamma}{16} \tag{B.12}$$

hold for all $t \leq t'$ if $p = \Omega(\frac{Md\log(d/\delta)}{\gamma})$ and $S_\lambda = O(\frac{\gamma\sqrt{\gamma p}}{\eta\sqrt{nM}})$. Furthermore, since $\|\frac{1}{p}W(0)W(0)^{\mathsf{T}} - I_d\| \leq \epsilon_0$ with high probability when $p = \Omega(d)$, we have

$$\|\frac{1}{p}XW(0)^{\mathsf{T}}W(0)X^{\mathsf{T}} - \gamma I_d\| \leq \|\frac{1}{p}XW(0)^{\mathsf{T}}W(0)X^{\mathsf{T}} - XX^{\mathsf{T}}\| + \|XX^{\mathsf{T}} - \gamma I_d\|$$

$$\leq (1 + \epsilon)\epsilon_0\gamma + \epsilon\gamma \leq \frac{\gamma}{16} \tag{B.13}$$

Therefore, combining (B.10), (B.11), (B.12) and (B.3), we have

$$\|e(t'+1)\| \leq \left(1 - \eta\lambda(t') - \eta\gamma\right)\|e(t')\| + \eta\left\|\frac{\eta}{p}XW(0)^{\mathsf{T}}W(0)X^{\mathsf{T}} - \gamma I_d\right\|\|e(t')\|$$

$$+ \eta(\|J_1(t')\| + \|J_2(t')\| + \|J_3(t')\|)\|e(t')\| + \eta\lambda(t')\|y\|$$

$$\leq \left(1 - \eta\lambda(t') - \eta\gamma\right)\|e(t')\| + \frac{1}{16}\eta\gamma\|e(t')\| + \frac{3}{16}\eta\gamma\|e(t')\| + \eta\lambda(t')\|y\|$$

$$\leq \left(1 - \eta\lambda(t') - \frac{\eta\gamma}{2}\right)\|e(t')\| + \eta\lambda(t')\|y\|$$

which completes the proof. $\qquad\square$

*Proof of Proposition 4.5.* By Corollary C.2, if $d = \Omega(\frac{1}{\epsilon}\log\frac{n}{\delta} + \frac{n}{\epsilon}\log\frac{1}{\epsilon})$, we have

$$\|XX^{\mathsf{T}} - I_n\| \leq \epsilon$$

It follows that $\lambda_{\min}(XX^{\mathsf{T}}) \geq 1 - \epsilon$ and $\lambda_{\max}(XX^{\mathsf{T}}) \leq 1 + \epsilon \leq (1 + 4\epsilon)(1 - \epsilon)$ for $\epsilon < 1/2$. $\quad\square$

**Lemma B.3.** *Recall from Lemma B.1 that*

$$\beta(t) = \prod_{i=0}^{t-1}(1 - \eta\lambda(i))\beta(0) - \frac{\eta}{\sqrt{p}}W(0)X^{\mathsf{T}}\hat{s}(t-1) + \frac{\eta^2}{p}b\hat{S}(t-1)$$

*with* $\hat{s}(t) = \sum_{i=0}^{t}\prod_{i<k\leq t}(1 - \eta\lambda(k))e(i)$ *and* $\hat{S}(t) = \sum_{i=0}^{t}\prod_{i<k\leq t}(1 - \eta\lambda(k))e(i)^{\mathsf{T}}XX^{\mathsf{T}}\sum_{j=0}^{i-1}e(j)$. *Under the conditions of Theorem 4.6, if* $t > C_1\frac{\log(p/\eta)}{\eta\lambda}$ *and* $\hat{S}(t) \geq \max(C_2\frac{\sqrt{p\gamma}}{\eta}\|\hat{s}(t)\|, 1)$ *for some positive constants $C_1$ and $C_2$, then* $\cos\angle(b, \beta(t)) \geq c$ *for some constant* $c = c_\delta$.

*Proof.* We compute the cosine of the angle between $\beta(t)$ and $b$. With probability $1 - \delta$,

$$\cos\angle(b, \beta(t)) = \frac{b^{\mathsf{T}}\beta(t)}{\|b\|\|\beta(t)\|} = \frac{\frac{b}{\|b\|}^{\mathsf{T}}\beta(t)}{\|\beta(t)\|}$$

$$\geq \frac{\frac{\eta^2}{p}\|b\|\hat{S}(t-1) - (1 - \eta\lambda)^t\|\beta(0)\| - \frac{\eta}{\sqrt{p}}\|\frac{b}{\|b\|}^{\mathsf{T}}W(0)\|\|X\|\|\hat{s}(t-1)\|}{\frac{\eta^2}{p}\|b\|\hat{S}(t-1) + (1 - \eta\lambda)^t\|\beta(0)\| + \frac{\eta}{\sqrt{p}}\|W(0)\|\|X\|\|\hat{s}(t-1)\|}$$

$$\geq \frac{c_1'\frac{\eta^2}{\sqrt{p}}\hat{S}(t-1) - c_2'\sqrt{p}(1 - \eta\lambda)^t - c_3'\eta\sqrt{\frac{d\gamma}{p}}\|\hat{s}(t-1)\|}{c_1'\frac{\eta^2}{\sqrt{p}}\hat{S}(t-1) + c_2'\sqrt{p}(1 - \eta\lambda)^t + c_4'\eta\sqrt{\gamma}\|\hat{s}(t-1)\|}$$

where we use (B.8), (B.9) and the tail bound for standard Gaussian vectors, and $c_i'$ are constants that only depend on $\delta$. Notice that if $t = \Omega(\frac{\log(p/\eta)}{\eta\lambda})$, we have $c_2'\sqrt{p}(1 - \eta\lambda)^t = O(\frac{\eta^2}{\sqrt{p}})$. It follows that $\cos\angle(b, \beta(t)) \geq c$ if $\hat{S}(t-1) = \Omega(\frac{\sqrt{p\gamma}}{\eta}\|\hat{s}(t-1)\| + 1)$. $\qquad\square$

**Lemma B.4.** *Consider the orthogonal decomposition $e(t) = a(t)\bar{y} + \xi(t)$, where $\bar{y} = -y/\|y\|$ and $\xi(t) \perp y$. Under the conditions of Theorem 4.6, there exists a constant $C_\tau > 0$ such that for any $t \in [\tau, T]$ with $\tau = \frac{C_\tau}{\eta\lambda}$, we have*

$$a(t) \geq \frac{\lambda - \gamma}{\lambda + \gamma}\|y\| \tag{B.14}$$

*and*

$$\|\xi(t)\| \leq \frac{\gamma}{\lambda + \gamma}\|y\|. \tag{B.15}$$

*Proof.* By Theorem 4.3, we have for all $t \leq T$, $\|e(t)\| \leq (1 - \eta\lambda - \eta\gamma/2)\|e(t)\| + \eta\lambda\|y\|$. By rearranging the terms, we have

$$\|e(t+1)\| - \frac{\lambda}{\lambda - \gamma/2}\|y\| \leq (1 - \eta\lambda - \frac{\eta\gamma}{2})\Big(\|e(t)\| - \frac{\lambda}{\lambda - \gamma/2}\|y\|\Big)$$

or

$$\|e(t)\| - \frac{\lambda}{\lambda - \gamma/2}\|y\| \leq (1 - \eta\lambda - \frac{\eta\gamma}{2})^t\Big(\|e_0\| - \frac{\lambda}{\lambda - \gamma/2}\|y\|\Big) \leq (1 - \eta\lambda)^t(\|e_0\| + \|y\|).$$

Notice that $\|y\|$ and $\|e(0)\|$ are of the same order, so when $t \in [\tau_1, T]$ with $\tau_1 = \frac{c_1}{\eta\lambda}$ and some constant $c_1$, we have

$$\|e(t)\| \leq \frac{\lambda + \gamma/2}{\lambda - \gamma/2}\|y\|. \tag{B.16}$$

In order to get a lower bound for $a(t)$, we multiply $\bar{y}^{\mathsf{T}}$ on both sides of (B.3). It follows that for $t \in [\tau_1, T]$

$$a(t+1) \geq \bar{y}^{\mathsf{T}}\Big(1 - \eta\lambda - \eta\gamma\Big)e(t) - \eta\|\frac{1}{p}XW(0)^{\mathsf{T}}W(0)X^{\mathsf{T}} - \gamma I_d\|\|e(t)\|$$

$$- \eta(\|J_1(t)\| + \|J_2(t)\| + \|J_3(t)\|)\|e(t)\| + \eta\lambda\|y\|$$

$$\geq (1 - \eta\lambda - \eta\gamma)a(t) - \frac{1}{4}\eta\gamma\|e(t)\| + \eta\lambda\|y\|$$

$$\geq (1 - \eta\lambda - \eta\gamma)a(t) + \frac{1}{2}\eta\gamma\|y\|.$$

In the second inequality, we use the bounds (B.10), (B.11), (B.12) and (B.13). The last inequality is by (B.16) and $\lambda \geq 3\gamma$. Following a similar derivation, we have

$$a(t) - \frac{\lambda - \gamma/2}{\lambda + \gamma}\|y\| \geq (1 - \eta\lambda - \eta\gamma)^{t-\tau_1}\left(a(\tau_1) - \frac{\lambda - \gamma/2}{\lambda + \gamma}\|y\|\right) \geq -(1 - \eta\lambda)^{t-\tau_1}(\|e(\tau_1)\| + \|y\|).$$

The bound (B.14) holds when $t \in [\tau_1 + \tau_2, T]$ with $\tau_2 = \frac{c_2}{\eta\lambda}$ and some constant $c_2$. Then we multiply $\frac{\xi(t+1)^\mathsf{T}}{\|\xi(t+1)\|}$ on both sides of (B.3). This establishes that for $t \in [\tau_1, T]$

$$
\begin{aligned}
\|\xi(t+1)\| &\leq \frac{\xi(t+1)^\mathsf{T}}{\|\xi(t+1)\|}\left(1 - \eta\lambda - \eta\gamma\right)e(t) + \eta\|\frac{1}{p}XW(0)^\mathsf{T}W(0)X^\mathsf{T} - \gamma I_d\|\|e(t)\| \\
&\quad + \eta(\|J_1(t)\| + \|J_2(t)\| + \|J_3(t)\|)\|e(t)\| + \eta\lambda\|y\| \\
&\leq (1 - \eta\lambda - \eta\gamma)\|\xi(t)\| + \frac{\eta\gamma}{4}\|e(t)\| \\
&\leq (1 - \eta\lambda - \eta\gamma)\|\xi(t)\| + \frac{\eta\gamma}{2}\eta\gamma\|y\|.
\end{aligned}
$$

The first inequality is by $\xi(t+1)^\mathsf{T}y = 0$ and in the second inequality we use $\xi(t+1)^\mathsf{T}e(t) = \xi(t+1)^\mathsf{T}\xi(t) \leq \|\xi(t+1)\|\|\xi(t)\|$. It follows that

$$\|\xi(t)\| - \frac{\gamma/2}{\lambda+\gamma}\|y\| \leq (1 - \eta\lambda - \eta\gamma)^{t-\tau_1}\left(\|\xi(0)\| - \frac{\gamma/2}{\lambda+\gamma}\|y\|\right) \leq (1 - \eta\lambda)^{t-\tau_1}(\|e(\tau_1)\| + \|y\|).$$

The bound (B.15) holds when $t \in [\tau_1 + \tau_3, T]$ with $\tau_3 = \frac{c_3}{\eta\lambda}$ for a constant $c_3$. Finally, the bounds (B.14) and (B.15) hold when $t \in [\tau, T]$ with $\tau = \tau_1 + \max(\tau_2, \tau_3)$. $\square$

**Lemma B.5.** *Under the conditions of Theorem 4.6, suppose* $T = \lfloor\frac{S_\lambda}{\lambda}\rfloor = C_T\frac{\sqrt{p}}{\eta\sqrt{n\gamma}}$. *Then we have* $\hat{S}(T) \geq \tilde{c}\frac{\sqrt{p\gamma}}{\eta}\|\hat{s}(T)\|$, *where* $C_T$ *and* $\tilde{c}$ *are positive constants.*

*Proof.* Notice that

$$e(i)^\mathsf{T}XX^\mathsf{T}e(j) \geq \gamma e(i)^\mathsf{T}e(j) - \|e(i)\|\|e(j)\|\|XX^\mathsf{T} - \gamma I\| \geq \gamma e(i)^\mathsf{T}e(j) - \epsilon\gamma\|e(i)\|\|e(j)\|.$$

For $i \in [T/2, T]$ and $\tau$ defined in Lemma B.4, we have

$$
\begin{aligned}
e(i)^\mathsf{T}XX^\mathsf{T}\sum_{j<i}e(j) &= e(i)^\mathsf{T}XX^\mathsf{T}\sum_{\tau\leq j<i}e(j) + e(i)^\mathsf{T}XX^\mathsf{T}\sum_{j<\tau}e(j) \\
&\geq \sum_{\tau\leq j<i}\left(\gamma e(i)^\mathsf{T}e(j) - \epsilon\gamma\|e(i)\|\|e(j)\|\right) - 2\gamma\sum_{j<\tau}\|e(i)\|\|e(j)\| \\
&\geq \sum_{\tau\leq j<i}\gamma\left(a(i)a(j) - \|\xi(i)\|\|\xi(j)\| - \epsilon\|e(i)\|\|e(j)\|\right) - 2c\tau\gamma\|y\|^2 \\
&\geq (i-\tau)\gamma\left[\left(\frac{\lambda-\gamma}{\lambda+\gamma}\right)^2\|y\|^2 - \left(\frac{\gamma}{\lambda+\gamma}\right)^2\|y\|^2 - \epsilon\left(\frac{\lambda+\gamma/2}{\lambda-\gamma/2}\right)^2\|y\|^2 - \frac{2c\tau}{i-\tau}\|y\|^2\right] \\
&\geq \frac{T}{8}\gamma\|y\|^2 = \frac{C_T}{8}\frac{\sqrt{p}}{\eta\sqrt{n\gamma}}\gamma\|y\|^2 \\
&\geq c\frac{\sqrt{p\gamma}}{\eta}\|y\|.
\end{aligned}
$$

$$(B.17)$$

The second inequality is the orthogonal decomposition of $e(i)$ and $\|e(i)\| \leq c\|y\|$ given by (4.3). The third inequality is by (B.14), (B.15) and (B.16) from Lemma B.4. The fourth inequality is by $\lambda = \Omega(\gamma)$, $i - \tau \geq T/4$ and the fact that $\tau/(i-\tau)$ is small ($p = \Omega(n)$). The last inequality is by

$\|y\| = \Theta(\sqrt{n})$. Therefore,

$$\hat{S}(T) = \sum_{i=0}^{T}(1-\eta\lambda)^{T-i}e(i)^\intercal XX^\intercal \sum_{j<i}e(j)$$

$$= \sum_{i=T/2}^{T}(1-\eta\lambda)^{T-i}e(i)^\intercal XX^\intercal \sum_{j<i}e(j) + (1-\eta\lambda)^{T/2}\sum_{i=0}^{T/2}(1-\eta\lambda)^{T/2-i}e(i)^\intercal XX^\intercal \sum_{j<i}e(j)$$

$$\geq \sum_{i=T/2}^{T}(1-\eta\lambda)^{T-i}c\frac{\sqrt{p\gamma}}{\eta}\|y\| + (1-\eta\lambda)^{T/2}\sum_{i=0}^{T/2}(1-\eta\lambda)^{T/2-i}c'T\gamma\|y\|^2$$

$$\geq \frac{c}{2}\frac{\sqrt{p\gamma}}{\eta}\frac{\|y\|}{\eta\lambda} - (1-\eta\lambda)^{T/2}\frac{c'T\gamma\|y\|^2}{\eta\lambda}$$

$$\geq \frac{c}{4}\frac{\sqrt{p\gamma}}{\eta}\frac{\|y\|}{\eta\lambda}$$

where the last inequality is by $(1-\eta\lambda)^{T/2} \ll 1$ when $p = \Omega(n)$. On the other hand,

$$\|\hat{s}(T)\| \leq \sum_{i=0}^{T}(1-\eta\lambda)^{T-i}\|e(i)\| \leq \frac{c}{\eta\lambda}\|y\|.$$

Combining the above inequalities gives the proof. $\qquad\square$

*Proof of Theorem 4.6.* First, notice that $\lambda(t) = 0$ when $t > T$. By Theorem 4.3 we have that the prediction error converges to zero exponentially fast, or $\|e(t+1)\| \leq (1-\eta\gamma/2)\|e(t)\|$. It follows that $\hat{S}(t) \to \hat{S}(\infty)$ and $\hat{s}(t) \to \hat{s}(\infty)$ as $t \to \infty$. By Lemma B.3, we know it suffices to show $\hat{S}(\infty) \geq C\frac{\sqrt{p\gamma}}{\eta}\|\hat{s}(\infty)\|$ with some constant $C$. Since

$$\hat{S}(\infty) = \sum_{i=0}^{\infty}(1-\eta\lambda)^{(T-i)_+}e(i)^\intercal XX^\intercal \sum_{j<i}e(j) = \hat{S}(T) + \sum_{i>T}e(i)^\intercal XX^\intercal \sum_{j<i}e(j)$$

and

$$\hat{s}(\infty) = \sum_{i=0}^{\infty}(1-\eta\lambda)^{(T-i)_+}e(i) = \hat{s}(T) + \sum_{i>T}e(i),$$

by Lemma B.5, it suffices to show

$$\sum_{i>T}e(i)^\intercal XX^\intercal \sum_{j<i}e(j) \geq C\frac{\sqrt{p\gamma}}{\eta}\sum_{i>T}\|e(i)\|. \tag{B.18}$$

We write $g = XX^\intercal \sum_{j<T}e(j)$. Then we have

$$\|g\| \geq \lambda_{\min}(XX^\intercal)\Big[\Big\|\sum_{\tau\geq j<T}e(j)\Big\| - \sum_{j<\tau}\|e(j)\|\Big]$$

$$\geq \lambda_{\min}(XX^\intercal)\Big[\sum_{\tau\geq j<T}a(j) - \sum_{j<\tau}\|e(j)\|\Big] \tag{B.19}$$

$$\geq \gamma\Big[(T-\tau)\Big(\frac{\lambda-\gamma}{\lambda+\gamma}\Big)\|y\| - \tau c\|y\|\Big]$$

and

$$\|g\| \leq \|XX^\intercal\|\Big(\sum_{j<\tau}\|e(j)\| + \sum_{\tau\geq j<T}\|e(j)\|\Big)$$

$$\leq (1+\epsilon)\gamma\Big[\tau c\|y\| + (T-\tau)\Big(\frac{\lambda+\gamma/2}{\lambda-\gamma/2}\Big)\|y\|\Big] \tag{B.20}$$

where we use the bounds (B.14) and (B.16) from Lemma B.4. We further denote $\alpha(t) = \bar{g}^\mathsf{T} e(t)$ where $\bar{g} = g/\|g\|$. Following the same calculation in (B.17), we have

$$
\begin{aligned}
g^\mathsf{T} e(T) &= e(T)^\mathsf{T} X X^\mathsf{T} \sum_{j<T} e(j) \\
&\geq (T-\tau)\gamma\Big[\Big(\frac{\lambda-\gamma}{\lambda+\gamma}\Big)^2 \|y\|^2 - \Big(\frac{\gamma}{\lambda+\gamma}\Big)^2 \|y\|^2 - \epsilon\Big(\frac{\lambda+\gamma/2}{\lambda-\gamma/2}\Big)^2 \|y\|^2 - \frac{2c\tau}{T-\tau}\|y\|^2\Big].
\end{aligned}
$$

Then

$$
\begin{aligned}
\frac{\alpha(T)}{\|e(T)\|} &\geq \frac{g^\mathsf{T} e(T)}{\|g\|\|e(T)\|} \\
&\geq \frac{(T-\tau)\gamma\Big[\Big(\frac{\lambda-\gamma}{\lambda+\gamma}\Big)^2 \|y\|^2 - \Big(\frac{\gamma}{\lambda+\gamma}\Big)^2 \|y\|^2 - \epsilon\Big(\frac{\lambda+\gamma/2}{\lambda-\gamma/2}\Big)^2 \|y\|^2 - \frac{2c\tau}{T-\tau}\|y\|^2\Big]}{(1+\epsilon)\gamma\Big[\tau c\|y\| + (T-\tau)\Big(\frac{\lambda+\gamma/2}{\lambda-\gamma/2}\Big)\|y\|\Big] \times \Big(\frac{\lambda+\gamma/2}{\lambda-\gamma/2}\Big)\|y\|} \\
&\geq \frac{\Big[\Big(\frac{\lambda-\gamma}{\lambda+\gamma}\Big)^2 - \Big(\frac{\gamma}{\lambda+\gamma}\Big)^2 - \epsilon\Big(\frac{\lambda+\gamma/2}{\lambda-\gamma/2}\Big)^2 - \frac{2c\tau}{T-\tau}\Big]}{(1+\epsilon)\Big[\frac{\tau c}{T-\tau} + \Big(\frac{\lambda+\gamma/2}{\lambda-\gamma/2}\Big)\Big] \times \Big(\frac{\lambda+\gamma/2}{\lambda-\gamma/2}\Big)}.
\end{aligned}
$$

Notice that $T/\tau = \Omega(\sqrt{p/n})$, so that when $p/n$, $\lambda/\gamma$ are large and $\epsilon$ is small, we have

$$
\alpha(T) \geq \frac{3}{4}\|e(T)\|. \tag{B.21}
$$

In order to obtain the lower bound on $\alpha(t)$ for all $t \geq T$, we multiply $\bar{g}^\mathsf{T}$ on both sides of (B.3). Notice $\lambda(t) = 0$ and apply the bounds (B.10), (B.11), (B.12) and (B.13). We have that

$$
\begin{aligned}
\alpha(t+1) &\geq (1-\eta\gamma)\bar{g}^\mathsf{T} e(t) - \eta\|\frac{1}{p}XW(0)^\mathsf{T} W(0)X^\mathsf{T} - \gamma I_d\|\|e(t)\| \\
&\quad - \eta(\|J_1(t)\| + \|J_2(t)\| + \|J_3(t)\|)\|e(t)\| \\
&\geq (1-\eta\gamma)\alpha(t) - \frac{\eta\gamma}{4}\|e(t)\|
\end{aligned}
$$

or for $t \geq T$,

$$
\alpha(t) \geq (1-\eta\gamma)^{t-T}\alpha(T) - \frac{\eta\gamma}{4}\sum_{i=T}^{t-1}(1-\eta\gamma)^{t-i}\|e(i)\|. \tag{B.22}
$$

Taking the sum over $t > T$, we have

$$
\begin{aligned}
\sum_{t>T}\alpha(t) &\geq \sum_{t>T}(1-\eta\gamma)^{t-T}\alpha(T) - \frac{\eta\gamma}{4}\sum_{t>T}\sum_{i=T}^{t-1}(1-\eta\gamma)^{t-i}\|e(i)\| \\
&\geq \frac{1-\eta\gamma}{\eta\gamma}\alpha(T) - \frac{\eta\gamma}{4}\sum_{i>T}\|e(i)\|\sum_{t>i}(1-\eta\gamma)^{t-i} \\
&\geq \frac{1-\eta\gamma}{\eta\gamma}\Big(\alpha(T) - \frac{\eta\gamma}{4}\sum_{i>T}\|e(i)\|\Big) \\
&\geq \frac{1-\eta\gamma}{\eta\gamma}\Big(\alpha(T) - \frac{1}{2}\|e(T)\|\Big) \\
&\geq \frac{1-\eta\gamma}{4\eta\gamma}\|e(T)\|.
\end{aligned} \tag{B.23}
$$

The second inequality follows from switching the order of sums. The fourth inequality is by exponential convergence after $T$ steps. The last inequality is by (B.21). With the above inequalities, we

are ready to bound the left hand side of (B.18), obtaining

$$
\begin{aligned}
\sum_{i>T} e(i)^\mathsf{T} X X^\mathsf{T} \sum_{j<i} e(j) &= \sum_{i>T} e(i)^\mathsf{T} X X^\mathsf{T} \sum_{j<T} e(j) + \sum_{i>T} e(i)^\mathsf{T} X X^\mathsf{T} \sum_{j\geq T} e(j) \\
&\geq \sum_{t>T} \alpha(t)\|g\| - 2\gamma \Big( \sum_{i\geq t} \|e(i)\| \Big)^2 \\
&\geq \frac{1-\eta\gamma}{4\eta\gamma} \|e(T)\| \gamma \Big[ (T-\tau)\Big(\frac{\lambda-\gamma}{\lambda+\gamma}\Big)\|y\| - \tau c\|y\| \Big] - 2\gamma\frac{4}{\eta^2\gamma^2}\|e(T)\|^2 \\
&\geq \frac{1-\eta\gamma}{4\eta\gamma} \|e(T)\| \gamma \Big[ (T-\tau)\Big(\frac{\lambda-\gamma}{\lambda+\gamma}\Big)\|y\| - \tau c\|y\| - \frac{64}{\eta\gamma(1-\eta\gamma)}\|y\| \Big] \\
&\geq \frac{1-\eta\gamma}{4\eta\gamma} \|e(T)\| \gamma \frac{T}{2} \|y\| = \frac{1-\eta\gamma}{4\eta\gamma} \|e(T)\| \gamma \frac{C_T}{2} \frac{\sqrt{p}}{\eta\sqrt{n}\gamma} \|y\| \\
&\geq C \frac{1-\eta\gamma}{4\eta\gamma} \frac{\sqrt{p\gamma}}{\eta} \|e(T)\|.
\end{aligned}
$$

(B.24)

The second inequality is by (B.23) and (B.19). The third inequality is by $\|e(T)\| \leq 2\|y\|$. The last inequality is by $\|y\| = \Theta(\sqrt{n})$. On the other hand,

$$
\sum_{i>T} \|e(i)\| \leq \sum_{i>T} (1-\eta\gamma/2)^{i-T}\|e(T)\| = \frac{1-\eta\gamma/2}{\eta\gamma/2}\|e(T)\|
$$

(B.25)

Combining (B.24) and (B.25) implies (B.18), as desired. $\qquad\square$

## C  Technical Lemmas

In this section, we list technical lemmas that are used in our proofs, with references. The first is a variant of the Restricted Isometry Property that bounds the spectral norm of a random Gaussian matrix around 1 with high probability.

**Lemma C.1** (Hand & Voroninski, 2018). *Let $A \in \mathbb{R}^{m\times n}$ has i.i.d. $\mathcal{N}(0, 1/m)$ entries. Fix $0 < \varepsilon < 1$, $k < m$, and a subspace $T \subseteq \mathbb{R}^n$ of dimension $k$, then there exists universal constants $c_1$ and $\gamma_1$, such that with probability at least $1 - (c_1/\varepsilon)^k e^{-\gamma_1 \varepsilon m}$,*

$$
(1-\varepsilon)\|v\|_2^2 \leq \|Av\|_2^2 \leq (1+\varepsilon)\|v\|_2^2, \quad \forall v \in T.
$$

Let us take $k = n$ in Lemma C.1 to get the following corollary.

**Corollary C.2.** *Let $A \in \mathbb{R}^{m\times n}$ has i.i.d. $\mathcal{N}(0, 1/m)$ entries. For any $0 < \varepsilon < 1$, there exists universal constants $c_2$ and $\gamma_2$, such that with probability at least $1 - (c_2/\varepsilon)^d e^{-\gamma_2 \varepsilon m}$,*

$$
\|A^\mathsf{T} A - I_m\| \leq \varepsilon
$$

Then following lemma gives tail bounds for $\chi^2$ random variables.

**Lemma C.3** (Laurent & Massart, 2000). *Suppose $X \sim \chi_p^2$, then for all $t \geq 0$ it holds*

$$
\mathbb{P}\{X - p \geq 2\sqrt{pt} + 2t\} \leq e^{-t}
$$

*and*

$$
\mathbb{P}\{X - p \leq -2\sqrt{pt}\} \leq e^{-t}.
$$

For two independent random Gaussian vectors, their inner product can be controlled with the following tail bound.

**Lemma C.4** (Gao & Lafferty, 2020). *Let $X, Y \in \mathbb{R}^p$ be independent random Gaussian vectors where $X_r \sim \mathcal{N}(0,1)$ and $Y_r \sim \mathcal{N}(0,1)$ for all $r \in [p]$, then it holds*

$$
\mathbb{P}(|X^\mathsf{T} Y| \geq \sqrt{2pt} + 2t) \leq 2e^t.
$$