# OpenReview forum: "Convergence and Alignment of Gradient Descent with Random Backpropagation Weights"
_NeurIPS.cc/2021/Conference — NeurIPS 2021 Poster_

### Official Review · Reviewer_B3Jb · 2021-06-29

**Rating:** 7
**Confidence:** 5

**Summary:**

This paper gives a theoretical analysis of feedback alignment (FA), an algorithm
to train neural networks by approximating the gradient of the loss function
using random matrices. More precisely, the weights of the network that is being
trained are replaced with random matrices in the back-propagation step.

The authors study two-layer neural networks in the over-parametrised regime,
where the number of samples n is much larger than the number of neurons in the
hidden layer, n, and provide two results:

1. A proof of linear convergence of the loss to 0. This proof follows the proofs
   of Du et al. (2018), Gao & Lafferty (2020) for standard backprop. The added
   difficulty of analysing DFA in this setting is that the effective kernel of
   the network is not a priori positive semi-definite.

2. An analysis of the alignment between the second-layer weights of the network
   and the feedback matrix used in the feedback alignment algorithm, which
   highlights the important role of regularisation and finds that (the role of)
   alignment is more complex than previously thought.


**Limitations And Societal Impact:**

- The authors clearly address the range of validity of their theoretical claims.
- Since this is a theoretical work, there are no immediate societal impacts that need to be addressed imho.

**Main Review:**

# Strengths

- The topic is timely - there has been increasing interest in feedback alignment
  methods and alternatives to back-prop more broadly, as evidenced by the
  NeurIPS 2020 workshop on this very topic and a number of recent papers (see below). However, an understanding of the
  power of these alternatives to backprop remains an open problem, so I see a
  need for theoretical work in this direction, which the paper addresses.
- The authors give clear and convincing mathematical arguments for two key
  problems of FA, namely convergence despite training on a surrogate loss, and
  alignment of the weights to the feedback vectors.
- The authors clarify the mechanics of alignment between weights and feedback
  matrix, which has been conjectured to be key to the success of the method.
- The paper is well-written: the exposition is clear, and I found that the
   main arguments were explained clearly.

# Weaknesses

While I found the paper insightful and the results interesting, the discussion
in its present version ignores several papers
that have recently provided analysis of feedback alignment algorithms. A discussion of the
following two theoretical papers, and how the results presented here relate to their results,
seems particularly appropriate:

- Regarding alignment, Frenkel et al. [Fre19] showed that using only the error sign is sufficient to
  maintain feedback alignment and to provide learning in the hidden layers of
  *linear* networks.
- Refinetti et al. [Ref21] analysed the dynamics of FA and DFA for two-layer
  non-linear networks in the feature learning regime.

Furthermore,
- the definition of alignment that the authors use (Def. 4.1) has been discussed
  extensively in the literature, cf. [Cra19, Fre19, Ref21]. This should be
  acknowledged.
- ...it might be worth mentioning that despite the problem with convolutions that
  the authors mention, DFA has recently been used to train modern neural
  networks such as Transformers [Lau20]

Let me also note that an important variant of feedback alignment is the direct feedback alignment of
  Nokland [Nok16], where the error vector is injected into each of the layers
  directly using a random matrix. For two-layer neural networks, the resulting
  update equations are the same, so the authors might want to mention that their results
  cover this relevant case, too.

# Some small comments

- What does the \tilde O notation mean? (l. 94)
- In the definition of the NTK after l. 137, the average <.> is taken w.r.t. to
  which distribution?

# References

- [Cra19] Crafton et al, Frontiers in neuroscience (2019). http://arxiv.org/abs/1903.02083
- [Fre19] Frenkel et al, (2019) http://arxiv.org/abs/1909.01311
- [Lau20] Launay et al. (2020) http://arxiv.org/abs/2006.12878
- [Nok16] Nøkland, NeurIPS 2016. http://arxiv.org/abs/1609.01596
- [Ref21] Refinetti et al, ICML 2021. https://arxiv.org/abs/2011.12428


**Time Spent Reviewing:**

4

---

> ### Author Response · Authors · 2021-08-09
> **Response**
>
> We thank the reviewer for compiling a list of references with relevant topics, which we will be sure to incorporate into the final version of our paper. For a detailed discussion of the connections between our results and previous work, please refer to our response to reviewer giv6.
>
> Please note that we use $\widetilde O(\cdot)$ to denote  the usual
> $O(\cdot)$ scaling while hiding logarithmic factors.
>
> In the definition of NTK after l.137, we use $\langle \cdot,\cdot\rangle$ to denote the inner product between two gradient vectors, and no expectation is taken in this equation. Given a kernel function $K(x,y)$, the inner product $\langle{f}, {g}\rangle_K = {\mathbb E}_{x,y}[f(x)^T K(x,x') g(x')]$
> has expectation with respect to the empirical distribution on a finite dataset.

---

> > ### Comment · Reviewer_B3Jb · 2021-08-17
> > **Thank you for your response**
> >
> > In light of the response of the authors, while also taking into account the comments of the other reviewers, I will keep my score at 7. I think this is a good paper, and a clear accept.

---

### Official Review · Reviewer_pXMq · 2021-07-13

**Rating:** 6
**Confidence:** 4

**Summary:**

This paper builds upon the work of Lillicrap et al 2016 of attempting to find biologically plausible methods for implementing backpropagation. They thoroughly describe the issues with non-local information and why there are limits to using backprop-based learning to understand more about biological neural systems.  They describe the algorithm of feedback alignment using random back-propagated weights.  The authors go on to confirm some of the results of the Lillicrap paper, and find a surprising and novel result that regularization is required to produce convergence in over-parameterized settings.
The experimental section of the paper shows some work on a more standard ML benchmark task of MNIST classification.  They show the results that regularization is required to get good performance on the task.


**Limitations And Societal Impact:**

The authors have not addressed this issue directly. They highlighted that a better understanding of biologically plausible algorithms will lead to a better understanding of the brain.  I don't believe there are immediate negative consequences of this work, but the authors could choose to include a statement making this explicit.

**Main Review:**

Overall, I believe this area of research, finding biologically plausible mechanisms, is an important direction for the field.  This will not only help us understand biological systems better, but also aid in the development of technologies that are more scalable and efficient.  The Lillicrap work was the introduction of a new concept, and this paper attempts to explain why and when the concept of random BP weights applies.  However, I don’t feel the authors really did enough in this paper to cross the threshold of a minimum publishable unit of work.  The main result of showing that regularization is required for convergence is an interesting result, but there is no theoretical basis as to why this is happening.  I’m ok with empirical results, but they haven’t demonstrated that this technique can really work on larger problems either.  Most of the paper is devoted to explaining the Lillicrap work and providing a notation.  In my view, either more empirical results or more well-motivated concepts are required to publish this work.

The quality of the writing is excellent, and the background provides a good platform to build from.  The explanations are clear and concise.  The experimental work is a bit sparse.  For instance, in figure 3 there is no discussion on why the classification performance on MNIST is best with lambda of 0.1 rather than 0.3.  How does the alignment relate to classification performance?  Why might the performance drop when alignment clearly increases?

The significance of the work is the main issue I have.  If the significance of the concepts presented were of much great magnitude, I would be ok with the level of rigor in the paper.  However, with an extension/clarification of a concept, there must an explanation of what new questions the work opens up.


**Time Spent Reviewing:**

4 hours

---

> ### Author Response · Authors · 2021-08-09
> **Response**
>
> We are grateful for your time and effort to review of our work.
> We would like to reply by emphasizing our contributions as well as addressing some of the concerns raised by the reviewer.
>
> We remark that our results do not include any statement implying that regularization is required for convergence.
> We show that regularization is needed for alignment, not convergence. We prove (under suitable conditions) that the algorithm converges in the sense that the error goes to zero exponentially fast---irrespective of alignment.
> We prove in the linear case that regularization is sufficient for alignment.
> Specifically, we show that the cosine of the angle between the random weights and the actual second layer weights will be bounded away from zero.
>
> Regarding the experiments, the reviewer comments that "in figure 3 there is no discussion on why the classification performance on MNIST is best with lambda of 0.1 rather than 0.3. How does the alignment relate to classification performance? Why might the performance drop when alignment clearly increases?"
> The fact that the accuracy is the worst with no regularization simply means that the model is overfitting. With too much penalization ($\lambda = 0.3$), the accuracy also suffers. This is just the usual bias-variance tradeoff. But the first plot of Figure 3 shows that, consistent with our analysis, the alignment increases with increasing regularization.

---

> ### Comment · Reviewer_pXMq · 2021-08-17
> **Changed score from 4->6**
>
> The minimum publishable unit is somewhat subject.  After reading all reviews, I decided to increase my score from 4->6 as others feel this paper's contribution is valuable.  As I said, the work is presented clearly and the paper is well written.  I still have the critique that the experimental results are thin and would prefer more to be done there to show the potential real-world impact of the work.

---

### Official Review · Reviewer_4iby · 2021-07-16

**Rating:** 8
**Confidence:** 3

**Summary:**

The paper provides a theoretical analysis of the feedback alignment algorithm (FA), a biologically plausible approximation to backpropagation. The authors consider a two-layered network, and derive training error bounds and error alignment for a variety of cases, using the neural tangent kernel method. The results show that FA-trained network can achieve zero training error for infinitely wide networks, and have some (strictly positive) degree of alignment between backprop and FA error vectors when proper regularization is used.

**Limitations And Societal Impact:**

The authors have adequately addressed the limitations and potential negative societal impact of their work.

**Main Review:**

Originality: The paper leverages the theory used for neural tangent kernels, but adapts it in a non-trivial way for feedback alignment. The results presented in the paper are new, and the related work is adequately cited.

Quality: The paper is technically sound.

Clarity: Overall, the paper is well written. I would suggest explaining spectral properties of $G(0)$ and $H(0)$ (around assumption 3.1) in more detail. One option is to mention the result of lemma A.3, and explain intuitively why $G(0)$ has large eigenvalues, and $H(0)$ -- small, as it is important for the whole method. I also disagree with the last line of the abstract, which says “with performance that is comparable with the full non-local backpropagation algorithm” -- we know that FA doesn’t perform well on hard tasks (see Akrout et.al. 2019 paper cited in this work).

Significance: The theoretical results of this paper are important for our understanding of feedback alignment. The results are somewhat limited as we know that FA fails on hard tasks (see Akrout et.al. 2019 mentioned above), but the presented method (and the NTK framework in general) might be useful for studying other alternatives to backprop.


**Time Spent Reviewing:**

3

---

> ### Author Response · Authors · 2021-08-09
> **Response**
>
> Thank you for your time and careful review of our work, and the positive feedback.
>
> Regarding the spectral properties of both $G(0)$ and $H(0)$,
> first note that $G(0) = UU^T$ is a positive semi-definite matrix, where $U = \frac{1}{\sqrt p}\psi(XW(0)^T)$, $X = (x_1,...,x_n)^T$ is $n$-by-$d$, and $W(0)$ is $p$-by-$d$. Intuitively, when $n\ll d\ll p$, the matrix $U$ will have full row rank and $G(0)$ becomes strictly positive definite. As for $H(0)$, each entry $H(0)_{ij} = \frac{1}{p}\sum_r h_r(i,j)$ is the sample mean
> of iid random variables $h_r(i,j)= \beta_r(0) b_r\psi'(w_r(0) x_i)\psi'(w_r(0) x_j)$.
>
> Since $\mathbb{E}(h_r(i,j)) = 0$, each $H(0)_{i,j}$ is close to zero, and so $\|\|H(0)\|\|$ is also close to zero. Lemma A.3 gives a rigorous proof based on these intuitions. We will try to present this more clearly in the final paper.
>
> We also thank the reviewer for pointing to the existing work that shows inferior performance of FA in certain settings. We can remove the last sentence in the abstract, since the previous empirical results are more mixed.

---

> > ### Comment · Reviewer_4iby · 2021-08-17
> > **Comment**
> >
> > Thank you for the response! I'm happy with the answer, and I'm keeping the current score of 8.

---

### Official Review · Reviewer_9xDE · 2021-07-17

**Rating:** 7
**Confidence:** 3

**Summary:**

The paper uses analysis techniques from Neural Tangent Kernels to prove that the feedback alignment algorithm converges in neural networks with a single non-linear hidden layer. In addition, they prove that the forward weights do not converge to alignment with the random backward weights.



**Limitations And Societal Impact:**

The authors adequately address this issue.

**Main Review:**

This is a well-written paper that appears to make a significant contribution to the understanding of the feedback aliignment / random backpropagation. Previous work has shown that this algorithm converges for special cases (deep chains of linear neurons, etc.), but this proof adresses a much more general case. However, I was not able to check the proof due to time constraints.

The second result about the weights failing to align is less surprising, but still interesting. A recent ICML paper by Refinetti, et al. "Align, then memorise: the dynamics of learning with feedback alignment" provides an analysis that is complementary to the results presented here.

EDIT after author rebuttal: The reviewers all agree that this is a well-written paper. I maintain that the contribution is significant enough for acceptance.

**Time Spent Reviewing:**

1

---

> ### Author Response · Authors · 2021-08-09
> **Response**
>
> Thank you for the positive feedback. The work of Renfinetti et al. provides a number of insights about feedback alignment and direct feedback alignment, especially showing that the learning dynamics follows two stages, where an alignment phase is followed by second phase that gives fast convergence of the error. As you observe, this analysis is complementary to our results. The key difference with our setting is that Renfinetti et al. assume in the two-layer case that the input dimension $d$ (in our notation) increases to infinity while the sample size $n$ and number of neurons $p$ are of order one. Our results show that the alignment phase does not happen in the high dimensional NTK regime where $p \gg d \gg n$, even while the algorithm's error converges to zero. We will expand of connections with this previous literature in the final version of our manuscript.

---

### Official Review · Reviewer_giv6 · 2021-07-19

**Rating:** 7
**Confidence:** 4

**Summary:**

This paper studies the convergence of Feedback Alignment, and proves that in the over-parameterized setting the error converges to zero exponentially fast. This is made possible by taking inspirations from recent work on the NTK, albeit with specific challenges as some quantities are not obviously positive semi-definite. Furthermore, it makes the somewhat surprising finding that regularization helps alignment, proving this in linear networks, and confirming this finding in simulations on Gaussian data and MNIST.

**Limitations And Societal Impact:**

Yes, the authors accurately describe the limitations and potential impact of their work.


**Main Review:**

*For ease of answering, I have annotated my various points with O.1/Q.1/etc...*

This is a solid and very well written paper, with a clear contribution to the field of alternative training methods.

The paper could be better positioned in the existing literature, in particular by mentioning all the works that has been done around Direct Feedback Alignment, such as Refinetti et al., 2020. Moreover, I think the surprising finding around regularization & alignment should be discussed more in depth.

Accordingly, **this paper currently stands marginally above the acceptance threshold (6)**. If the two points above were to be addressed, I would be willing to increase my score to a clear accept (8).

### Originality

The paper is an interesting application of NTK methods to an open-problem in alternative training methods, the convergence of Feedback Alignment. It is well placed in the immediate literature, but misses a number of important papers on the theory of such methods.

**O.1**: Notably, the authors completely ignore the large body of literature around Direct Feedback Alignment (Nøkland, 2016). DFA is an extension of FA, where the random projection of the error is directly sent to each layer, enabling parallelization of weight updates after the loss is calculated. In particular, Refinetti et al., 2020 have done an extensive analysis of the dynamics of convergence & alignment in DFA. This analysis brings a number of important findings around alignment, which it would be interesting to see discussed and compared with the work here done. Furthermore, Frenkel et al. 2019 have also provided a theoretical analysis for a variant of DFA, Direct Random Target Projection. To go a bit further, this could also lead to bridging the gap between some of the theoretical findings made here and the real-world behaviors of FA/DFA: open questions remain, such as FA/DFA failing on convolutions (Bartunov et al., 2018) but working in many other architectures (Launay et al., 2020). A discussion of the impacts of the findings made in this paper in terms of our general understanding of FA/DFA would be very valuable.

### Quality

The theoretical contributions are sound and well backed. The experiments to confirm the findings around regularization are interesting, although I think the authors do not go in depth enough in their analysis.

**Q.1**: In results both on synthetic data and MNIST (Figure 2 & Figure 3), it is interesting to see that despite lower alignment in non-regularized nets, end-task performance is not necessarily improved by higher alignment values (the lambda = 0.0 of Figure 3 probably overfits). This warrants a more in depth discussion. Could the higher alignment values be an artifact of the weights being pushed closer to zero? (This is noted by the authors l185.) This would motivate disentangling the components of alignment due to L2 regularization, and the "true alignment" of the raw gradients. This would be really interesting to measure and plot, and could provide further elements to understand this "surprise" of regularization helping alignment. The authors also briefly mention experiments with dropout, highlighting that a study of other regularization methods (dropout, etc.) could also prove interesting.

### Clarity

The paper is well written and the mathematical notations clear. The structure is also good, making it an enjoyable paper to read.

### Significance

The findings of the authors are interesting, and contribute to a broader understanding of feedback alignment methods.

I do feel that the addition of a discussion of the potential impacts of these findings on the current understand of FA/DFA, as well as a more in depth discussion around the "surprising" regularization finding would help make this paper more impactful for the community.

**Time Spent Reviewing:**

3

---

> ### Author Response · Authors · 2021-08-09
> **Response**
>
> Thank you for your review and the thoughtful comments on our work. You ask for two improvements to the paper (1) more discussion of related work and (2) further discussion of our finding about alignment and regularization.
>
> *Discussion of related work*
>
> We have not been fully "plugged in" to this recent literature, and are very happy to learn of these references. The direct feedback alignment procedure is simple and elegant, and allows for greater parallelization; for two-layer networks, it is the same as the algorithm that we analyze. The empirical results of Launay et al. are remarkable. In our view the failure of the algorithm on convolutional networks is somewhat orthogonal, since the architecture itself is not biologically plausible. While the original Nøkland (2016) paper does not establish convergence, even in the linear case, the simplicity of DFA may allow for a more direct application of our analysis tools for the multilayer nonlinear setting. We look forward to taking a crack at this. The work of Renfinetti et al. provides a number of insights about feedback alignment and direct feedback alignment, especially showing that the learning dynamics follows two stages, where an alignment phase is followed by second phase that gives fast convergence of the error. The key difference with our setting is that Renfinetti et al. assume in the two-layer case that the input dimension $d$ (in our notation)  increases to infinity while the sample size $n$ and the number of neurons $p$ are of order one. Our results show that the alignment phase does not happen in the high dimensional NTK regime where $p \gg d \gg n$, even while the algorithm's error converges to zero. We will expand of connections with this previous literature in the final version of our manuscript.
>
> *Disentangling penalization and alignment*
>
>  The reviewer asks about the relationship between the "end-task performance" and the alignment based on results from Fig. 2 and Fig. 3. We should note that the actual meaning of the algorithm "performance" given by the loss (Fig. 2) and accuracy (Fig. 3) are different. Fig. 2 gives a comparison of convergence speed of the algorithm on the training set, and it shows that larger regularization $\lambda$ will slow down the convergence but improve alignment. This is consistent with our Theorem 4.2, which shows that regularization adds an extra term to the training error $e(t)$. However, the right panel of Fig. 3 shows the predicted accuracy on the *test set*, which reflects the generalization error of the algorithm. The result here is due to bias-variance trade-off: When there is no regularization, the model is over-fitting, but with large regularization, the estimation is more biased.
>
> In the review, the question "Could the higher alignment values be an artifact of the weights being pushed closer to zero?" is raised. As we suggest in the discussion section, the shrinkage of the initial weights is proposed to be the main reason why regularization improves alignment. To investigate this further, you make the nice suggestion to try to "disentangle the components of alignment due to L2 regularization, and the 'true alignment' of the raw gradients."  We discuss this briefly in Section 6,
> but we would like to explain more here. In the linear case, the second layer parameters $\beta(t)$ have a decomposition
> $$
> \beta(t) = (1-\eta\lambda)^t \beta(0) + a_\lambda(t) b + v_\lambda(t)
> $$
>
> and these three terms are almost orthogonal to each other. When there is no regularization ($\lambda=0$), the first term is the initial $\beta(0)$ and the second and third terms are from the gradient updates. Since the scale of the first term is dominant under over-parameterization (Theorem 3.2, Section 4.1), there is no alignment. When $\lambda > 0$, the first term converges to zero exponentially, but the second term, although it also shrinks a little, becomes substantial, so that alignment happens, with $\cos\angle(b,\beta(t)) > c$.
>
> We have extended our MNIST analysis to show how this plays out empirically.
> After adding the term $(1-\eta \lambda)^t \beta(0) - \beta(0)$ to the parameters
> $\beta(t)$ obtained without regularization, we see that alignment is significantly increased, while the accuracy drops. We will include a plot showing this in the final version of the paper.

---

> > ### Comment · Reviewer_giv6 · 2021-08-17
> > **Score increased to an accept**
> >
> > Thanks to the authors for addressing my comments. In light of their answer, I have increased my score to an **7 (accept).**
> >
> > > The key difference with our setting is that Renfinetti et al. assume in the two-layer case that the input dimension $d$ (in our notation) increases to infinity while the sample size $n$ and the number of neurons $p$ are of order one.
> >
> > As a small note: while the number of neurons in the teacher & student in Refinetti et al. is indeed O(1) and $d$ increases infinity, I do not believe that $n$ is explicitly taken to be of order one.

---

### Decision · Program_Chairs · 2021-09-27

**Decision:**

Accept (Poster)

**Comment:**

This paper studies the convergence of Feedback Alignment (FA), an alternative to backpropagation that has been found empirically to work well, but until very recently did not received any theoretical justification of any sort, as the weights of the network that is being trained are replaced with random matrices in the back-propagation step.

This paper is an interesting addition in the study of these “biologically”  plausible algorithm. The authors study two-layer neural networks in the over-parametrised regime, and mainly provide two results: a) A proof of linear convergence of the loss to zero error and b) An analysis of the alignment between the second-layer weights of the network and the feedback matrix used in the feedback alignment algorithm.

The consensus among the reviewer was that, despite FA not being a mainstream algorithm, given its interest as a potential solution to the weight transport problem, the study was of interest to the Neurips community, and that the paper provided a definitely worthy  contribution. Given the rigour and the level of detail in the paper, this work has been judged to provide as well a blueprint for theoretical analysis of other biologically plausible learning rules, and therefore has a potential to impact future research.

The interaction between the authors and the reviewers during the rebuttal phase have seen some reviewers increasing their grade for the paper. Currently, all reviewers are largely in favor of acceptance, and thus the area chair recommend acceptance as well.